# Immunoglobulin expression and the humoral immune response is regulated by the non-canonical poly(A) polymerase TENT5C

Aleksandra Bilska [1,9], Monika Kusio-Kobiałka[1,2,9], Paweł S. Krawczyk [2,3], Olga Gewartowska[2,3], Bartosz Tarkowski[2,3], Kamil Kobyłecki[2], Dominika Nowis [4,5], Jakub Golab [6,7], Jakub Gruchota[2,3], Ewa Borsuk[2,3,8], Andrzej Dziembowski [1,2,3 ✉] & Seweryn Mroczek [1,3 ✉]

TENT5C is a non-canonical cytoplasmic poly(A) polymerase highly expressed by activated B cells to suppress their proliferation. Here we measure the global distribution of poly(A) tail lengths in responsive B cells using a Nanopore direct RNA-sequencing approach, showing that TENT5C polyadenylates immunoglobulin mRNAs regulating their half-life and consequently steady-state levels. TENT5C is upregulated in differentiating plasma cells by innate signaling. Compared with wild-type, *Tent5c*[−/−] mice produce fewer antibodies and have diminished T-cell-independent immune response despite having more CD138[high] plasma cells as a consequence of accelerated differentiation. B cells from *Tent5c*[−/−] mice also have impaired capacity of the secretory pathway, with reduced ER volume and unfolded protein response. Importantly, these functions of TENT5C are dependent on its enzymatic activity as catalytic mutation knock-in mice display the same defect as *Tent5c*[−/−]. These findings define the role of the TENT5C enzyme in the humoral immune response.

---

[1] Institute of Genetics and Biotechnology, Faculty of Biology, University of Warsaw, Pawinskiego 5a, 02-106 Warsaw, Poland. [2] Institute of Biochemistry and Biophysics, Polish Academy of Sciences, Pawinskiego 5a, 02-106 Warsaw, Poland. [3] Laboratory of RNA Biology, International Institute of Molecular and Cell Biology, Trojdena 4, 02-109 Warsaw, Poland. [4] Genomic Medicine, Medical University of Warsaw, Banacha 1a, 02-097 Warsaw, Poland. [5] Laboratory of Experimental Medicine, Center of New Technologies, University of Warsaw, Banacha 2c, 02-097 Warsaw, Poland. [6] Department of Immunology, Medical University of Warsaw, Banacha 1a, 02-097 Warsaw, Poland. [7] Centre of Preclinical Research, Medical University of Warsaw, Banacha 1a, 02-097 Warsaw, Poland. [8] Department of Embryology, Institute of Zoology, Faculty of Biology, University of Warsaw, Miecznikowa 1, 02-096 Warsaw, Poland. [9] These authors contributed equally: Aleksandra Bilska, Monika Kusio-Kobiałka. ✉email: adziembowski@iimcb.gov.pl; seweryn.mroczek@gmail.com

The development of an adaptive humoral immune response requires the activation of resting B cells following antigen recognition. This process is associated with structural and functional changes leading to the generation of high-affinity memory B cells and antibody-secreting plasma cells (ASC). Extensive B cell differentiation is characterized by the clonal expansion, somatic hypermutation leading to affinity maturation, isotype switching, and formation of ASC or memory cells. At a cellular level, this process involves the reorganization of the rough endoplasmic reticulum (RER) and Golgi compartments to promote immunoglobulin (Ig) synthesis, assembly, and secretion[1,2]. These global physiological changes occurring during B cell activation and differentiation are linked to broad changes in the transcriptomic profile, which is controlled by the coordinated action of regulatory networks of transcriptional factors, such as NF-κB, BCL6, IRF4, and BLIMP1[3]. Studies have also shown the involvement of RNA-binding proteins and microRNAs in shaping the B cell transcriptome[4,5], suggesting that posttranscriptional gene expression regulation plays an important role in B cell physiology[6–9]. In addition to global transcript changes, B cell maturation is associated with a large increase in the translation of mRNAs targeted to the ER[1,10].

Almost every mRNA molecule, except histone mRNAs, is polyadenylated during 3′-end processing. Nuclear polyadenylation is mediated by canonical poly(A) polymerases that interact with 3′-end cleavage machinery. The poly(A) tail has a critical function in mRNA stability and translation efficacy as nearly all mRNA decay pathways begin with the removal of poly(A) tails[11–13]. Previous studies of polyadenylation in B cells have focused on alternative polyadenylation and splicing in the regulation of Ig class switching[14–17]. However, in addition to the polyadenylation that occurs in the nucleus, the poly(A) tail can be expanded in the cytoplasm by non-canonical poly(A) polymerases (ncPAPs). This process is considered to have an important function in the activation of dormant deadenylated mRNAs during gametogenesis[18] and in neuronal processes, but has not been studied in B cells. We and others have identified a metazoan-specific family of cytoplasmic poly(A) polymerases, TENT5 (previously known as FAM46)[19–21]. In mammals, this family has four members, among which TENT5C is the best characterized. The importance of TENT5C is underscored by the occurrence of *Tent5c* somatic mutations in ~20% of cases of multiple myeloma (MM) patients. Further work revealed that TENT5C is a bona fide MM cell growth suppressor[21,22]. TENT5C polyadenylates multiple mRNAs with a strong specificity to those encoding ER-targeted proteins. This partially explains TENT5C toxicity to MM cells since an increased protein load caused by the stabilization of ER-targeted mRNAs enhances the ER stress, to which MM is very sensitive. Initial characterization of *Tent5c* knockout (KO) in mice revealed that it might play a role in the physiology of normal B cells, since isolated primary splenocytes from *Tent5c* KO mice proliferate faster upon activation than those isolated from wild-type (WT) animals[21,22].

Here, we show the role of TENT5C in B cells in more detail. Using Nanopore direct RNA sequencing, we provide global analysis of poly(A) tail distribution in B cells from WT and *Tent5c* KO animals, and show that the primary targets of TENT5C are mRNAs encoding Igs. mRNAs encoding all classes of Ig have shorter poly(A) tails. Importantly, further studies indicate that the production of Igs is lower in *Tent5c* KO B cells, leading to decreased gamma globulin concentrations in KO mice serum and diminished humoral responses after immunization with thymus-independent (TI) antigens. TENT5C-deficient cells are characterized by accelerated growth rate and faster differentiation to CD138[high] plasma cells (PCs), which explains the increased number of these cells in the bone marrow (BM) and spleen of KO mice. Accordingly, TENT5C expression is limited to late stages of B cell lineage differentiation and is highly upregulated by innate signaling via specific Toll-like receptors (TLRs). Despite the acceleration of B cell proliferation rate, a lack of TENT5C results in a decreased ER compartment volume, reduced dynamics of its expansion during B cell activation, and down-regulation of unfolded protein response (UPR). All these phenotypes are reproduced in mice expressing catalytically dormant TENT5C (D90N, D92N), which confirms that they result from the enzymatic activity of this ncPAP. In aggregate, here we show that cytoplasmic polyadenylation by ncPAP TENT5C regulates the humoral immune response.

## Results

**Ig mRNAs are specific TENT5C targets**. TENT5C is implicated in the polyadenylation of mRNAs encoding proteins passing through the ER in MM cells, which originate from terminally differentiated B cells[21]. In order to identify TENT5C substrates in activated B cells, we implemented Oxford Nanopore Technologies (ONT) direct full-length RNA sequencing to measure poly(A) tail length at a genome-wide scale. Unlike traditional RNA-seq techniques, the Nanopore-based system detects DNA or RNA single molecules as they traverse through protein channels, without the need for an enzymatic synthesis reaction. Moreover, despite higher per-base error-rate, this sequencing strategy avoids limitations and biases introduced during the amplification of long homopolymers, such as adenine tracts within poly(A) tails as PCR amplification of cDNA is not required during library preparation[23,24] (Fig. 1a). In the case of RNA sequencing, the substrate is a whole single RNA molecule with the motor protein attached to its 3′-end, which passes the RNA strand through the pore at a consistent rate in an ATP-dependent manner. As the sequencing proceeds in the 3′ to 5′-direction, the adaptor oligo is detected first, followed by the poly(A) tail, then the entire body of the transcript is sequenced. For efficient sequencing, pure mRNA fractions are needed, and to avoid any biases total RNA was subjected to an mRNA enrichment step using the mutated recombinant elongation initiation factor 4E (GST-eIF4E[K119A]), which has a high affinity to 5′-cap structure[25,26] (Supplementary Fig. 1a, b). According to our experience and previous reports, this is the most effective strategy for mRNA enrichment[26], as the efficiency of mRNA enrichment and depletion of other unwanted high-abundance RNA species was estimated by quantitative PCR (qPCR) and northern blot analysis for selected transcripts (Supplementary Fig. 1c–e). RNA prepared in such a way isolated from lipopolysaccharide/interleukin 4 (LPS/IL-4)-activated splenic B cells (WT and *Tent5c* KO mice) was adapted for Nanopore direct RNA sequencing with the MinION device[24] (Fig. 1a).

Using ONT sequencing, we obtained in total 2.5 million mappable transcriptome-wide full-length native-strand mRNA reads (Supplementary Table 1), which were used for the analysis of steady-state poly(A) tail lengths in responsive B cells. The data were reproducible (Supplementary Fig. 1f) while the accuracy of poly(A) length estimations was evaluated with the poly(A) standards, barcoded mRNA transcripts with predefined poly(A) tails[27], introduced at library preparation step. Distribution of their poly(A) lengths was consistent with the previous reports, with peak values as expected (Supplementary Fig. 1g). We observed no global changes in the mRNA polyadenylation status between WT and TENT5C-deficient cells (Fig. 1b). However, mRNAs encoding Igs had their peak length of the poly(A) tails significantly decreased from 60 in WT to 41 adenosines in KO cells. This was observed for transcripts encoding both heavy and light Ig chains (Fig. 1c, d). The effect was highly specific to Ig transcripts as poly(A) tails of other highly abundant mRNAs, such as ribosomal

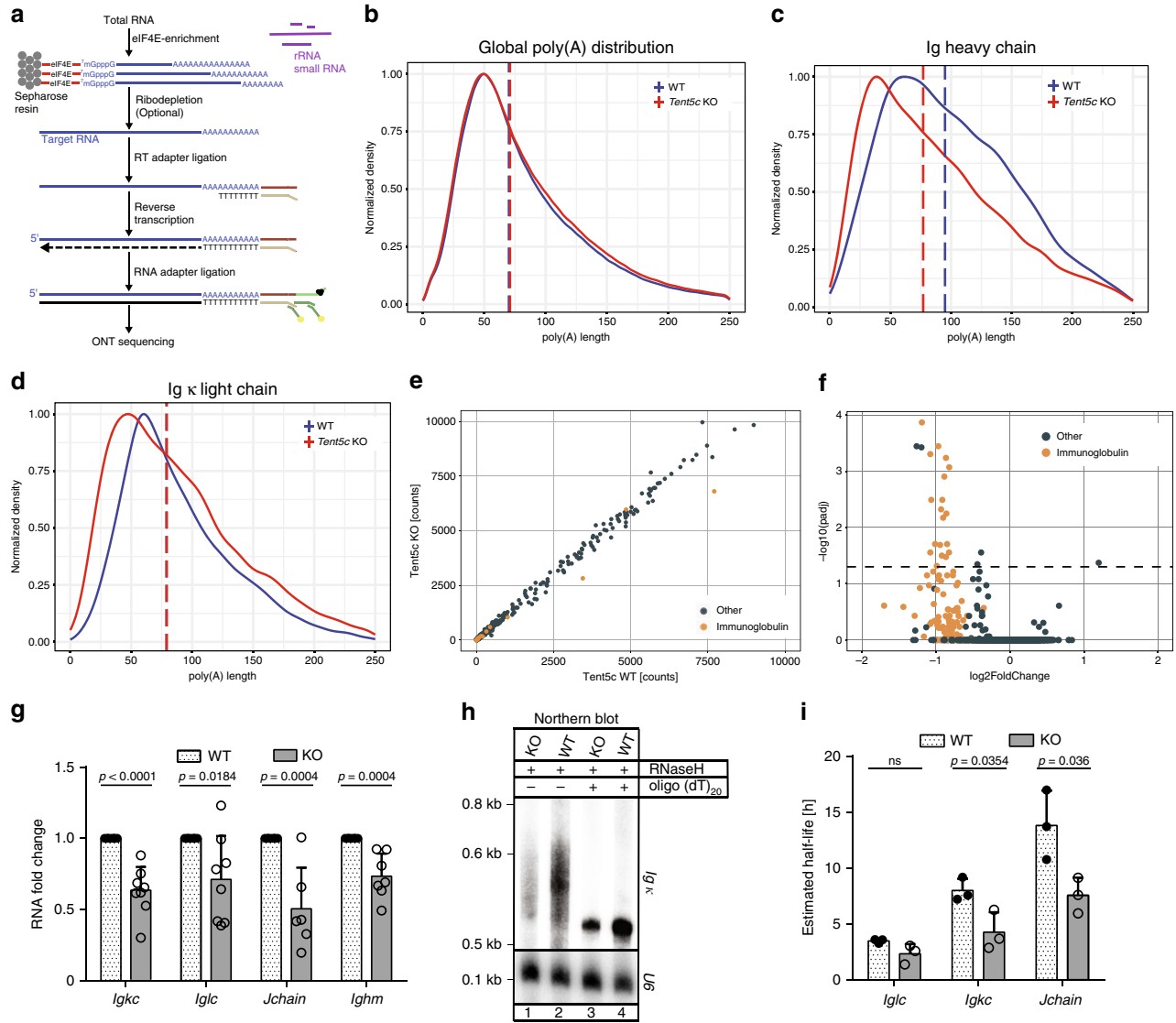

**Fig. 1 TENT5C regulates the steady-state level of Ig mRNAs through polyadenylation. a** Library preparation workflow for Nanopore direct RNA sequencing. **b–d** Nanopore-based poly(A) lengths profiling of mRNA isolated from WT and *Tent5c* KO B cells activated for 7 days. Density distribution plots for all transcripts **b**, Ig heavy **c**, and Ig light (kappa) chains transcripts **d** scaled to a maximum of 1, are shown. Vertical dashed lines represent median poly (A) lengths for each condition (for **b** and **d** lines for both conditions are overlapped). **e** Scatter plot of raw counts for individual transcripts obtained with Nanopore direct RNA sequencing for WT and *Tent5c* KO. Igs transcripts ($n = 39$) are marked in orange. **f** Volcano plot showing the results of DESeq2 differential expression analysis on data from Illumina sequencing of RNA isolated from B cells of WT and *Tent5c* KO mice activated for 7 days. Igs transcripts ($n = 138$) are marked in orange. Dashed line marks the *P* value significance threshold (0.05). **g** RT-qPCR analysis of Igs mRNA expression in WT and *Tent5c* KO cells. Bars represent mean fold change values ± SD (*Jchain* $n = 6$, *Ighm* $n = 7$, *Igkc*, *Iglc* $n = 8$). Values are shown as fold changes to WT. **h** The Ig κ mRNA poly(A) tail is reduced in *Tent5c* KO cells. Northern blot analysis of Ig κ mRNA and U6 RNA (loading control) from *Tent5c* KO and WT B cells activated for 7 days, treated with RNase H in the presence of oligo(dT)$_{20}$. **i** RNA half-life calculation with 4-thiouridine labeling method. WT and *Tent5c* KO B cells activated for 7 days were labeled with 4sU for 1 h. RNA was biotinylated and 4sU-labeled RNA was purified on streptavidin beads. Igs mRNAs half-lives were estimated based on 4sU-labeled RNA enrichment measured by RT-qPCR analysis. Bars represent the mean of calculated half-life values ± SD ($n = 3$). *P* values were calculated using DESeq2 (v.1.22) with standard settings **f**; two-tailed unpaired Student's *t*-test **g**, **i**; not significant (ns); *n*—biological replicates. Source data: Supplementary Datasets 1 (1b–e) and 2 (1f), Source Data file (1g, i), Supplementary Fig. 8 (1h).

proteins or mitochondrial transcripts were not affected at all (Supplementary Fig. 1h, i). Importantly, poly(A) tails of mitochondrial transcripts were predicted to be ~52 adenosines (Supplementary Fig. 1i), being in agreement with the existing knowledge[28–32]. Finally, differential expression analysis based on ONT data showed that the abundance of Ig mRNAs was also decreased in *Tent5c* KO cells (Fig. 1e, Supplementary Dataset 1).

In parallel with direct RNA-seq, we performed standard Illumina RNA sequencing, allowing us to perform a comparative correlation analysis between datasets. Again, Ig mRNAs were the most downregulated ones in *Tent5c* KO (Fig. 1f, Supplementary Dataset 2), which was additionally confirmed by RT-qPCR (Fig. 1g). To verify RNA-seq data on Ig expression, naive, spleen-derived B cells from WT and KO mice were activated with LPS and IL-4, and selected mRNAs were analyzed by northern blot. The Ig κ transcripts are again less abundant and migrate faster on the gel in *Tent5C* KO compared to those from WT, confirming that they are TENT5C substrates (Fig. 1h, Supplementary Fig. 1j). Next, we confirmed that the observed Ig-encoding mRNA lengthening is indeed caused by mRNA polyadenylation using

an RNase H cleavage assay with oligo(dT)$_{20}$ (Fig. 1h). Importantly, *Tent5c* is the only one of the four members of the *Tent5* gene family expressed at detectable levels in B cells, and undergoing strong induction during their activation and differentiation as we measured by real-time qPCR (Supplementary Fig. 1k). Interestingly, its expression positively correlates with the upregulation of *Pabpc1*, suggesting that mRNA polyadenylation contributes to transcriptional reprogramming of the B cell response (Supplementary Fig. 1k).

Finally, in order to determine whether the Ig mRNAs polyadenylation impacts their stability, we evaluated the half-lives of selected transcripts identified in ONT sequencing as TENT5C substrates. Using 4-thiouridine (4sU) labeling of nascent transcripts followed by biotinylation, separation on streptavidin beads, and qPCR-based estimations, a significant decrease in half-life of Ig mRNAs was detected in *Tent5c* KO cells (Fig. 1i). Experiments based on transcription inhibition by actinomycin D revealed the same trend (Supplementary Fig. 1l). Therefore, we conclude that mRNA polyadenylation by TENT5C leads to Igs mRNA stabilization and enhances their expression.

**B cells isolated from *Tent5c* KO produce fewer antibodies.** Next, we analyzed the impact of inefficient transcript polyadenylation on Ig production level using western blot. *Tent5c* KO B cells grown in vitro produced fewer antibodies than those isolated from WT littermates (both light and heavy chains), while the levels of other secreted proteins: IL-6, as well as ER-associated chaperonin—GPR94, were not changed as much (Fig. 2a). Accordingly, the analysis of media collected from B cell cultures confirmed decreased levels of secreted antibodies by activated TENT5C-deficient B lymphocytes (Fig. 2b). Moreover, flow cytometry revealed that the level of IgG1-positive cells is significantly reduced in TENT5C-deficient B cells after 3 days of activation with IL-4 and LPS in vitro (Fig. 2c).

Next, to get insight into physiological regulation of Ig production, we assessed the intracellular (cytoplasmic) levels of IgG1 and IgA in a population of CD138-positive cells isolated from the BM, and spleen of WT and *Tent5c* KO mice. The intracellular levels of Igs, in contrast to surface Igs, reflect the secreting ability of these cells. We observed that the percentages of IgG1 and IgA-positive cells are lowered in *Tent5c* KO mice, which confirms the altered Ig production (Fig. 2d, e). To extend these studies, we did blood serum protein electrophoresis (SPEP) to compare globulin fractions and ELISA tests to assess Ig concentrations in WT and KO mice serum. In agreement with the previous in vitro and in vivo cell analyses, we have found a substantial decrease of gamma globulin fraction (which contains mainly whole antibodies) in KO mice plasma reflecting alterations in their production and secretion by B cell lineage (Fig. 2f). The decrease in gamma globulins was specific since total serum protein concentrations, as well as albumin, alpha, and beta globulin plasma subfractions, were not reduced. A slight decrease of alpha 2 globulin concentration in *Tent5c* KO serum is most likely a consequence of previously reported by us microcytic anemia[21] that develops in KO mice presumably as a result of inhibited globin synthesis, but not due to iron uptake deficiency as its levels were unchanged in the serum (Fig. 2g). In agreement with this, quantification of IgA, IgM, IgG1, IgG2a, IgG2b, and IgG3 Igs in serum by ELISA revealed significantly decreased concentrations in KO mice compared to WT littermates (Fig. 2h).

Finally, to assess if the lack of TENT5C affects the development of humoral response, we immunized mice against T-cell-dependent (TD; ovalbumin (OVA)) and T-cell-independent TI (TNP-LPS) antigens. Quantification of specific antibodies by ELISA tests showed clearly that TI, but not TD response is

substantially affected in KO comparing to WT mice (Fig. 2i, j). This also suggests that polyadenylation by TENT5C plays a more important role in early innate immune response signaling rather than in T-dependent antibody responses.

**TENT5C is expressed at late stages of B cell differentiation.** Next, we have generated *Tent5c*-GFP knock-in mice using the CRISPR/Cas9 approach and used flow cytometry to analyze TENT5C expression in B cell lineage further. Similarly to the previously published *Tent5c*-FLAG[21], the GFP knock-in mouse line did not display any gross phenotype. In vitro-activated B cells isolated from *Tent5c*-GFP mice revealed a distinct GFP-positive cell population, absent in the non-tagged controls confirming the utility of our mouse model (Fig. 3a). Naive B cells from *Tent5c*-GFP mice and WT littermates were activated with LPS and IL-4, and subjected to cytometric analysis to measure TENT5C-GFP and CD138 PC markers levels in a time-dependent manner. This analysis revealed that TENT5C is mainly expressed in the population of CD138-positive cells, suggesting the involvement of this enzyme in the last steps of B cell differentiation (Fig. 3b). To confirm this hypothesis, we have systematically examined spleen and BM-residing B cell and PC subpopulations from young adult (12–15 weeks) unimmunized WT and *Tent5c*-GFP mice using multicolor flow cytometry. The GFP-tag knock-in does not affect the general distribution of B cell and PC populations. However, this approach revealed that the CD138$^{high}$ B cell subset is also highly GFP positive in both BM (up to 76%) and spleen (up to 93%; Fig. 3c). In turn, GFP-positive cells were not detected in B cell subsets at the early stages of differentiation (Supplementary Fig. 2). The detailed gating strategy is presented in Supplementary Figs. 3 and 4. Thus, TENT5C-GFP is mainly expressed in the last stages of B cell differentiation as revealed by detailed PC subpopulation analyses, including dividing plasmablasts (CD19$^{high}$/CD45R$^{high}$), early PCs (CD19$^{high}$/CD45R$^{low}$), and mature resting PCs (CD19$^{low}$/CD45R$^{low}$; Fig. 3d, Supplementary Fig. 4). Finally, we confirmed those results with immunostaining of spleen sections showing that GFP-fused TENT5C is mainly expressed in CD138-positive cells (Fig. 3e).

**TENT5C expression is stimulated by innate signaling.** Next, to define whether other stimuli than a combination of LPS and IL-4 lead to TENT5C upregulation, we have tested the main types of B cell activators. Naive B cells isolated from *Tent5c*-GFP mice and WT littermates as controls were activated with a panel of TLR receptor agonists, mainly pathogen-associated molecular patterns molecules, ligands of CD40 (TD signaling), and B cell receptor (BCR). Subsequent flow cytometry analyses of those cells revealed a significant number of GFP and CD138-positive cells, similar to positive control, as a result of stimulation of selected TLR receptors, including TLR1/2 (Pam3CSK4), TLR2 (HKLM), TLR4 (LPS, *Escherichia coli* K12), TLR6/2 (FSL1), and TLR9 (ODN1826) (Fig. 4). Stimulation of TLR3 (low and high molecular weight Poly(I:C)), TLR5 (Flagellin *Salmonella typhimurium*), and TLR8 (ssRNA40/LyoVec) showed a rather limited effect on TENT5C-GFP expression.

Subsequent analysis of the PC subpopulations in CD138$^{high}$ (Q3 for WT; Fig. 4a) and CD138$^{high}$GFP$^{pos}$ (Q2 for TENT5C-GFP; Fig. 4b) cell fractions, using CD19 and CD45R markers showed a similar response to the treatment for cells isolated from both mouse lines. In turn, signaling provided by BCR stimulated with polyclonal F(ab′)2 goat anti-mouse IgM and CD40 receptor with megaCD40L (trimeric variant) had limited effect on TENT5C expression; however, they induced the differentiation of PCs (Fig. 4).

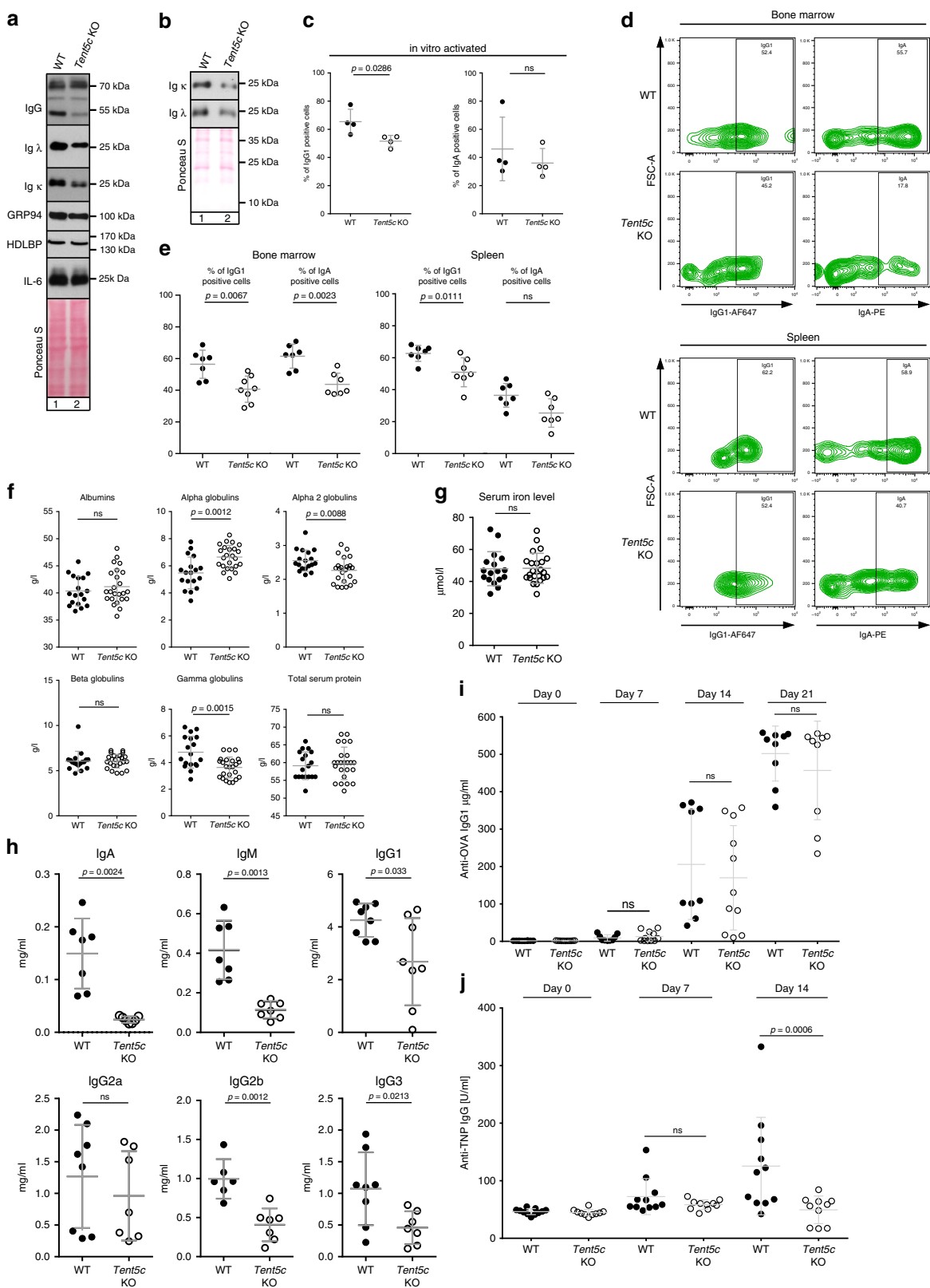

To assess whether antibody production is also impaired in *Tent5c* KO B cells activated via innate signaling pathways, we activated B cells from WT and KO with TLR agonists with the highest effect on TENT5C expression (TLR1/2, TLR4, TLR6/2, and TLR9 agonists) in a time course, then collected cells and medium for western blot analysis at days 3 and 5. Indeed, we observed decreased secretion of both heavy IgG and light chains in KO, while α-tubulin and GRP94 levels were unaffected (Supplementary Fig. 5).

All this together links B cell-intrinsic innate signaling via selected TLRs with the regulation of adaptive humoral immunity

**Fig. 2 *Tent5c* KO produces less antibodies and has impaired immune response. a, b** Western blot analysis of extracellular IgG heavy and light (κ and λ) chains, GRP94, HDLBP, and IL-6 **a** or secreted κ and λ light chains **b** from WT and *Tent5c* KO activated B cells. Ponceau S staining was used as a loading control. **c** Percentage of IgG1 and IgA-positive cells in WT and *Tent5c* KO B cells activated in vitro for 3 days (*n* = 4). **d** Intracellular Ig isotypes levels in mature resting PCs. Cell populations were analyzed based on CD138, CD19, CD45R, IgG1, and IgA staining. Pseudo-color dot blots are showing FSC-A vs IgG1-AF647 or IgA-PE fluorescence. Gating strategy in Supplementary Fig. 4. **e** Percentage of mature resting PC-positive cells for intracellular IgG1 or IgA isotypes (*n* = 7 or *n* = 8 for % of IgG1-positive cells in KO BM). **f** Blood serum albumins, alpha 1 globulins, alpha 2 globulins, beta globulins, gamma globulins, and total protein levels in *Tent5c* KO and WT by SPEP (WT *n* = 18, KO *n* = 24). **g** Iron levels in blood serum of *Tent5c* KO and control animals (WT *n* = 17, KO *n* = 23). **h** Measurements of IgA, IgM, IgG1, IgG2a, IgG2b, IgG3 in WT and *Tent5c* KO mice serum (*n* = 6–8). **i** TENT5C deficiency does not impair the immune response against a T-cell-dependent antigen. OVA-specific serum IgG1 antibodies concentrations were measured with ELISA on days 7, 14, and 21 postimmunization with OVA-CFA. Mean anti-OVA IgG1 concentrations (μg/ml) ± SE (WT *n* = 9, KO *n* = 10). **j** TENT5C deficiency impairs the immune response against a type 1 T-cell-independent antigen. TNP-specific serum IgG concentrations were measured with ELISA on days 7 and 14 postimmunization with TNP-LPS. The graph shows mean anti-TNP IgG concentrations (U/ml) ± SE (WT *n* = 11, KO *n* = 10). *P* values were calculated using two-tailed Mann–Whitney *U* test **c, e**; two-tailed unpaired Student's *t*-test with Welch's correction **f–h**; one-way ANOVA with Tukey's multiple comparison test **i, j**; not significant (ns); data shown as mean values ± SD, unless otherwise indicated; *n*—biological replicates. Source data: Supplementary Fig. 8 (2a, b) and Source Data file (2c, e–j).

modulated by ncPAP TENT5C and reveals this enzyme as an important modulator of B cell response.

**_Tent5c_ KO leads to accelerated B cell differentiation.** *Tent5c* KO leads to an increased B cell proliferation, suggesting that TENT5C may control the process of their differentiation into PCs[21]. Since the terminal differentiation of B cells takes place in the secondary lymphoid organs, we carried out extended phenotyping of B cells in the spleen, as well as BM. First, we observed that spleens in KO mice are particularly ~20% enlarged compared to those isolated from WT (Fig. 5a), while there is no difference in overall animal mass (Fig. 5b). As this observation strongly suggested enhanced proliferation rates, we have systematically examined B cell subpopulations from spleens and BM of conventionally cohoused adult unimmunized littermates (12–16 weeks) using multicolor flow cytometry.

Interestingly, we observed that the number of CD138[high] cells in the spleen and BM was significantly increased in *Tent5c* KO mice (Fig. 5c, d). Next, we carried out the quantitative determination of CD138[high] PC subpopulations in WT and KO mice. This has revealed mature resting PCs as the only cells whose number is increased in *Tent5c* KO mice spleen and BM, while the numbers of dividing plasmablasts and early PCs were slightly decreased or not changed in KO (Fig. 5e). Interestingly, other B cell subpopulations in the BM (pre-proB, pro-B, pre-B, immature, early/late mature, and transitional B) and in the spleen (transitional (T1/T2/T3), marginal zone (MZP and MZ), follicular (I and II)) were not affected by *Tent5c* KO (Supplementary Fig. 6). All these observations clearly suggest that the lack of TENT5C leads to the enhanced B cell proliferation and differentiation in vivo, and confirms previous in vitro findings.

Next, we asked whether *Tent5c* KO influences the secondary antibody repertoire generated by class switch recombination (CSR), which replaces IgM with other isotypes during B cell differentiation into PCs, and whether it differs in the KO compared to the WT. The changes in the class profile of presented antibodies may indicate a disturbance in the CSR process. We observed that CD138[high] B cell subsets in *Tent5c* KO lose IgA expression much faster as compared with WT, which confirms their faster proliferation and accelerated selection of IgG1-expressing polyclonal PCs, indicating alerted CSR in the KO mutant (Fig. 6a, b). The general concentration of Igs is lowered in KO cells as a result of a diminished expression level and they accumulate membrane-bound IgG1 (Fig. 6c, d). This is in agreement with the results, we obtained for in vitro cultured B cells (Fig. 2a, b).

**_Tent5c_ is essential for ER expansion in activated B cells.** The transition of B cells into Ig-secreting PCs requires a significant expansion of secretory organelles, given that ER-specific

chaperones and folding enzymes facilitate the posttranslational structural maturation of Ig. Since TENT5C modifies transcripts encoding Ig in responding B cells, it may have a possible ER-related function. To examine the link between TENT5C activity and ER functionality, we performed a fractionation of activated B cells isolated from *Tent5c*-GFP mice through discontinuous sucrose gradients followed by western blot analysis[33]. This approach confirmed that TENT5C-GFP mainly localized in the cytosol. However, there is a minor fraction of the membrane-bound enzyme that suggests its involvement in the mRNA polyadenylation directly on ER (Fig. 7a). It was supported by a partial intracellular co-localization of endogenous TENT5C-GFP protein with the ER selectively stained by tracker dye in ex vivo-activated B cells (Fig. 7b). Finally, co-immunoprecipitation (Co-IP) experiments using high-affinity anti-GFP nanobodies followed by high-resolution mass spectrometry (MS) revealed that TENT5C-GFP interacts with ribosomal proteins, suggesting that TENT5C which may directly polyadenylate Ig mRNAs at the RER (Supplementary Dataset 3).

As B lymphocyte maturation requires a significant increase in ER volume and *Tent5c* KO leads to the increased rate of the differentiation process, we compared its size and expansion dynamics during the activation of WT and KO B cells, using specific ER-tracker dye labeling followed by flow cytometry analyses. The results of these experiments showed that a lack of TENT5C impairs the capacity of the secretory pathway through the reduction of ER volume (naive and mature resting PC; Fig. 7c, d). Moreover, the dynamic of the ER expansion after activation is much slower in the TENT5C-deficient cells despite their accelerated differentiation into plasmocytes (Fig. 7e). Reduced ER in *Tent5c* KO cells is consistent with decreased levels of both Ig-encoding transcripts and the main chaperone Hspa5 (BIP), necessary for the correct functioning of the ER (Supplementary Dataset 2). Thus, our findings, together with the fact that the overall rate of antibody production in KO cells is reduced, strongly suggests that B cells isolated from *Tent5c* KO may have reduced ER stress levels. In order to analyze the ER stress response, we treated activated WT and KO B cells with the standard ER stress-inducing agent tunicamycin (Tu), and then analyzed UPR markers with qPCR and western blots. Interestingly, the induction of ER stress enhances *Tent5c* expression (Fig. 7f). Accordingly, the KO cell response for Tu treatment was significantly diminished as shown by *Xbp1* mRNA splicing and expression of selected markers *Ire1*, *Perk*, *Gro94*, *Chop*, and *Ero1-LB*, showing a general downregulation of UPR (Fig. 7g–i). In contrast, the initial ER stress level is enhanced in the mutant compared to WT, which probably reflects faster differentiation to PCs.

Concluding, TENT5C dysfunction leads to a reduced ER volume and capacity of the ER stress response, as a probable consequence of a decreased load of Ig.

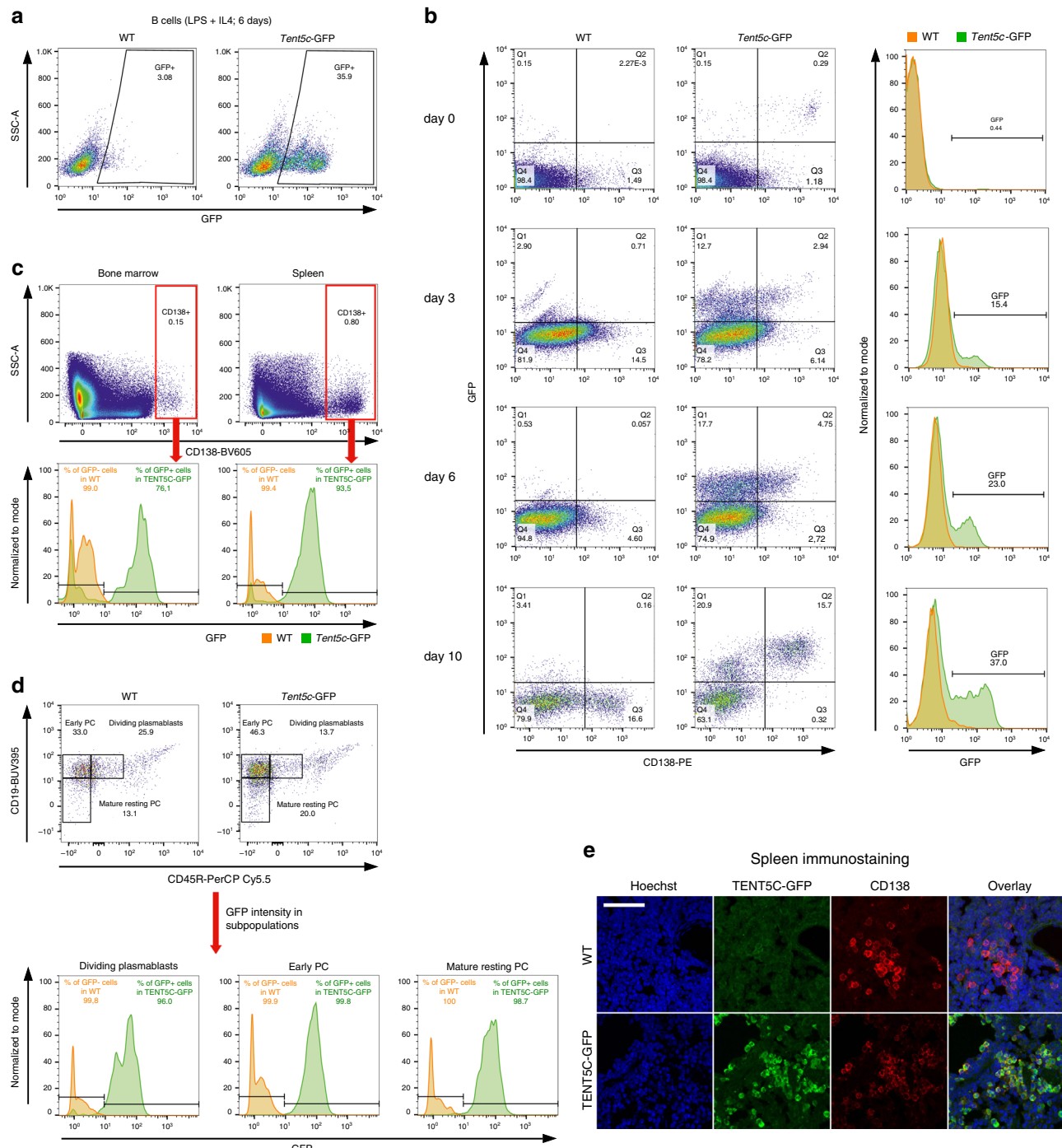

**Fig. 3 Expression of TENT5C is limited to the last stages of B cell lineage differentiation. a, b** Flow cytometry analysis of TENT5C-GFP expression in a subpopulation of CD138[high] cells activated in vitro with LPS and IL-4 for 6 days **a** or up to 10 days **b** presented as pseudo-color dot blot showing SSC-A vs GFP fluorescence or GFP vs CD138-PE fluorescence **a** and/or histograms of GFP-fluorescence intensity **b**. Orange color refers to WT, green is related to *Tent5c*-GFP. A detailed gating strategy is presented in Supplementary Fig. 4. **c, d** Flow cytometry analysis of TENT5C-GFP expression in splenocytes and BM **c** and different splenic PC subpopulations **d**: dividing plasmablasts, early PC, mature resting PC. Color code as above. Pseudo-color dot blots are showing SSC-A vs CD138-BV605 fluorescence or CD19-BUV395 vs CD45R-PerCP Cy5.5 fluorescence. See also Supplementary Figs. 2 and 4. **e** Immunohistochemical staining for PC marker CD138, and GFP in spleens of wild-type and *Tent5c*-GFP knock-in mice. Scale bar denotes 50 μm.

***Tent5c* catalytic mutant mice reproduce KO phenotypes.** Since our attempts to show direct interaction of TENT5C with mRNA substrates were unsuccessful, we wanted to see whether the observed phenotypes are directly dependent on the TENT5C activity. To this end, knock-in mice bearing TENT5C D90N and D92N mutations, which, as we have previously shown, inactivate the enzyme[21] have been constructed (herein described as *Tent5c*

Cat). Firstly, we have analyzed RNA isolated from activated B cells from Cat and WT littermates (Fig. 8a, Supplementary Fig. 1j), and we found Igs mRNA poly(A) tail shortening similar to that observed in cells isolated from KO mice (Fig. 1h). Subsequent western blot analysis of activated B cells, as well as the media from in vitro cultures, showed a specific reduction in the levels of Igs (Fig. 8b, c) and lowered percentages of cells positive for

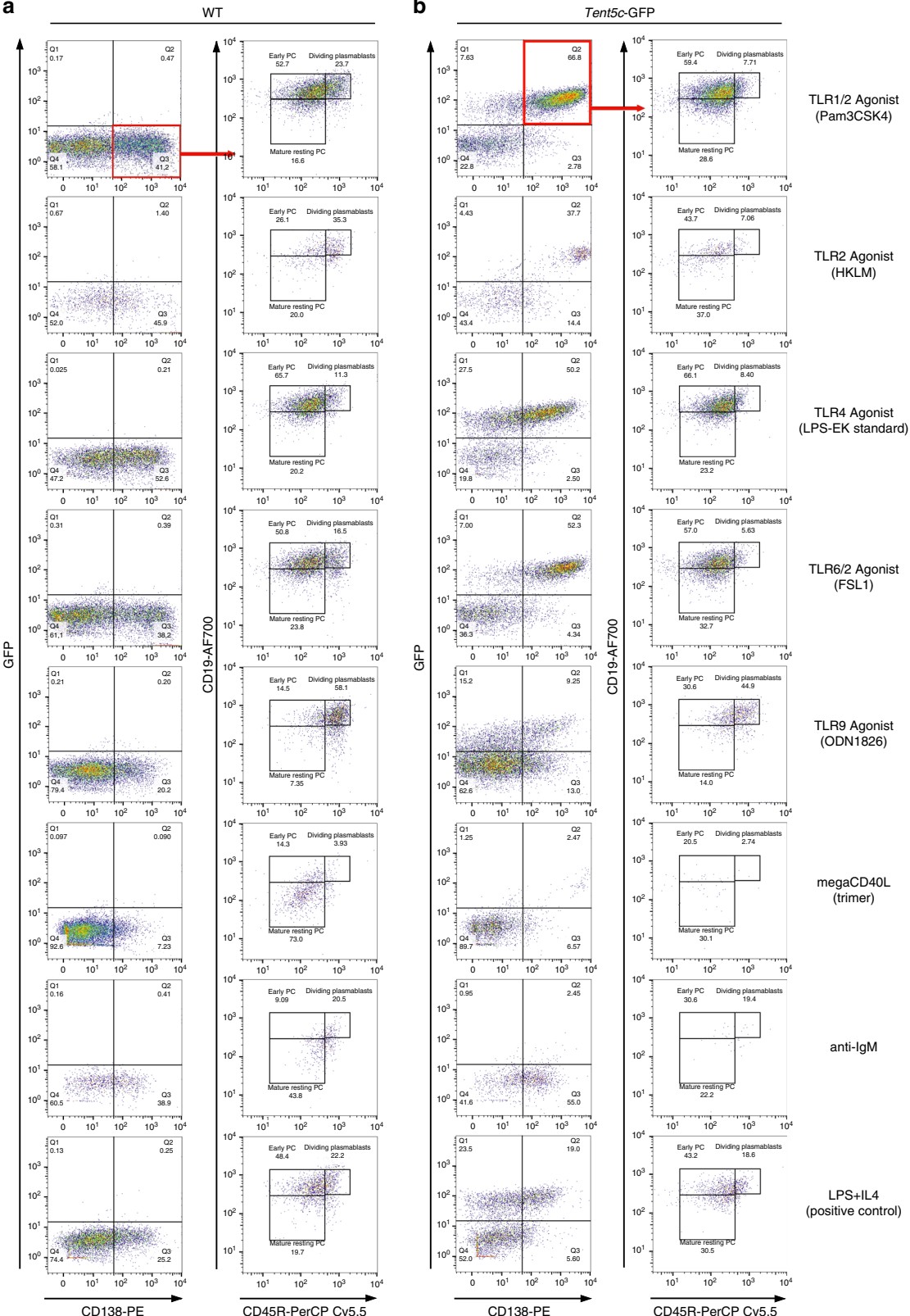

**Fig. 4 TENT5C is specifically upregulated by specific TLR. a**, **b** Analysis of the B cells activation process, measured by the evaluation of co-expression of CD138 and GFP molecules (red quarter, left columns), and further differentiation process of CD138^highGFP^pos into dividing plasmablasts, early PC, and mature resting PC (right column). B cells isolated from WT **a** and *Tent5c*-GFP **b** mice, were treated with different activators (indicated on the right part of the plot) for 4 days: TLR1/2 agonist (Pam3CSK4), TLR2 agonist (HKLM), TLR4 agonist (LPS-EK standard), TLR6/2 agonist (FSL1), TLR9 agonist (ODN1826), megaCD40L (trimer), anti-IgM, and LPS/IL-4 as a positive control. Flow cytometry analysis was based on the live/dead, CD138, CD45R, and CD19 staining and additionally GFP. Pseudo-color dot blots are showing GFP vs CD138-PE fluorescence or CD19-AF700 vs CD45R-PerCP Cy5.5 fluorescence. See also Supplementary Fig. 4.

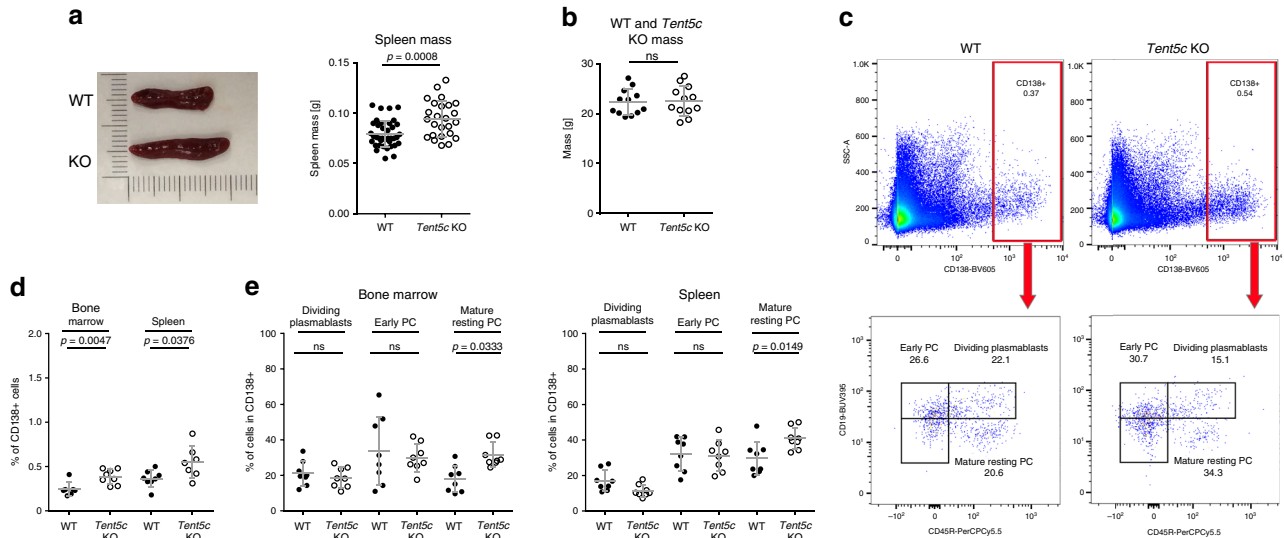

**Fig. 5 TENT5C negatively regulates B cell lineage differentiation into PCs. a** Comparison of spleen mass in WT and *Tent5c* KO mice (right panel; WT $n =$ 38, KO $n = 25$). Pictures (left panel) show the representative difference in the spleen size between WT and *Tent5c* KO. **b** Comparison of body mass in WT and *Tent5c* KO mice ($n = 12$). **c** Flow cytometry quantitative analysis of CD138[positive] cells (upper pseudo-color dot blots) and particular CD138[high] subsets: dividing plasmablasts, early PC, and mature resting PC (lower pseudo-color dot blots) in WT and *Tent5c* KO. PC populations were analyzed based on CD138, CD19, CD45R, IgM, IgG1, and IgA staining. Pseudo-color dot blots are showing SSC-A vs CD138-BV605 fluorescence or CD19-BUV395 vs CD45R-PerCP Cy5.5 fluorescence. The gating strategy is presented in Supplementary Fig. 4. **d** Percentages of CD138[high] cells in WT and *Tent5c* KO in both BM and spleen. (BM: WT $n = 7$, KO $n = 8$, spleen: WT $n = 8$, KO $n = 7$). See Supplementary Fig. 6 for other cell subpopulations. **e** Comparison of CD138[high] subpopulations (selected as shown in **c**): dividing plasmablasts, early PC, mature resting PC in WT, and *Tent5c* KO in both BM and spleen ($n = 8$). P values were calculated using two-tailed unpaired Student's *t*-test with Welch's correction **a**, **b**, two-tailed Mann–Whitney U test **d**, and two-way ANOVA with post hoc Bonferroni test **e**; not significant (ns); data are presented as mean values ± SD; $n$ — biological replicates. Source: Supplementary Fig. 8 (5a left) and and Source Data file (5a right, 5b, d, e).

intracellular IgG1 in a CD138[+] population (Fig. 8d, e) although, the expression of surface IgG1 isoform is enhanced (Fig. 8d, e). In agreement with all these results, analysis of serum proteinograms revealed decreased gamma globulins fraction (Fig. 8f), while other fractions do not differ much (Supplementary Fig. 7a).

At the organismal level, loss of TENT5C catalytic activity leads to the increased spleen size (Fig. 8g) and elevated overall CD138[+] cell numbers in BM and spleen (Fig. 8h). Detailed phenotyping of CD138[+] cells subpopulation from BM and spleen confirmed faster proliferation rate, resulting in an increased number of mature resting PC and decreased the number of cells at earlier steps of differentiation (dividing plasmablasts and early PC) in both BM (Fig. 8i) and spleen (Fig. 8j). Importantly, the early stages of B cell differentiation are not affected in *Tent5c* Cat mice (Supplementary Fig. 7b, c). In agreement with this naive B cells isolated from WT and *Tent5c* Cat activated in vitro with LPS and IL-4 accelerate differentiation into mature PC (Supplementary Fig. 7d).

Finally, similarly to the results obtained with cells isolated from KO, the dynamics of the ER expansion after activation is much slower in the *Tent5c* Cat cells (Fig. 8k) despite their accelerated differentiation into plasmocytes. All these results prove that described above Ig-related phenotypes in TENT5C-deficient mice are a direct effect of mRNA polyadenylation impairment rather than the absence of TENT5C protein.

## Discussion

B cell development in mice and humans has been extensively studied, revealing complex physiological changes, driven by different signaling pathways, which influence the genome (somatic hypermutation, CSR), transcriptome (through the coordinated action of transcription factors), and proteome (ER reorganization, posttranscriptional gene expression regulation) in differentiating cells. In this study, we provide evidence for cytoplasmic

polyadenylation, driven by TENT5C, being the previously undescribed mechanism involved in the regulation of Ig expression, which dysfunction leads to aberrant B cell differentiation. Our data indicate that the role of cytoplasmic polyadenylation is broader than previously anticipated and provides a new layer to the regulation of Ig expression.

Recently, we presented the first experimental data for a new family of non-canonical cytoplasmic poly(A) polymerases TENT5 (formerly FAM46)[34]. One of the members, TENT5C, was shown to be a specific growth suppressor in MM cells[21,22] and is the only TENT5 family member expressed at significant levels in B lymphocytes. In this work, using direct RNA sequencing by ONT, we identified Ig mRNAs as TENT5C specific targets in activated B cells. This experimental strategy offers high-quality, full-length mRNA sequences, including UTRs and poly(A) tails giving deeper insights into transcriptome shaping, comparing to classical RNA-seq experiments[32]. It is characterized by a relative technical simplicity, and in contrast to other RNA-seq methods, no PCR biases are introduced into libraries as in case of other currently used RNA 3′-end research techniques, such as Nascent RNAend-Seq, TAIL-seq, PAL-seq, TED-seq, PAC-Seq, EnD-seq, FLAM-seq, or most recently PAIso-seq[31,35–41]. Importantly, despite the recent dynamic expansion of RNA 3′-terminome research, little was known how cytoplasmic ncPAP enzymes contribute to gene expression programs since such techniques were never applied for KOs of individual enzymes in physiological conditions. Cytoplasmic adenylation was mostly studied in the context of gametogenesis or in other instances, in which transcription is arrested or spatially and temporarily separated. To our knowledge, this is the first report showing an in-depth insight of poly(A) tails dynamics and enzyme specifically controlling their length under native physiological conditions in responding B lymphocytes. Importantly, a shortening of Ig mRNAs poly(A) tail by ~30%

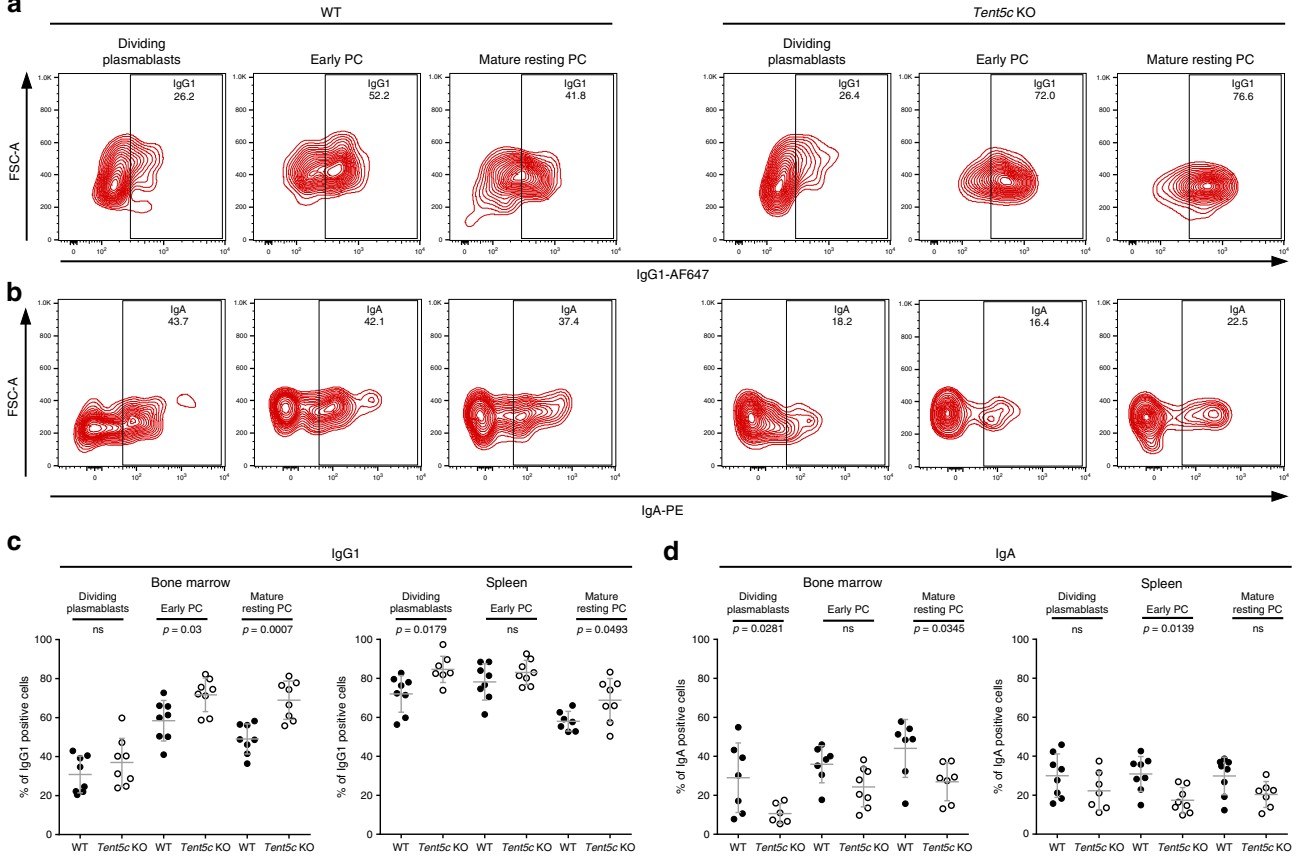

**Fig. 6 Tent5c KO plasmocytes are characterized by the abnormalities in the CSR. a, b** Flow cytometry analysis of surface IgG1 **a**, and IgA **b**, in dividing plasmablasts, early PC, and mature resting PCs isolated from BM or spleen (representative contour blots are shown). PC populations were analyzed based on CD138, CD19, CD45R, IgG1, and IgA staining. Pseudo-color dot blots are showing FSC-A vs IgG1-AF647 fluorescence or FSC-A vs IgA-PE fluorescence. See the gating strategy in Supplementary Fig. 4. **c, d** Percentage of cells positive for surface IgG1 **c**, or IgA **d**, Igs in dividing plasmablasts, early PC, and mature resting PCs isolated from BM or spleen. Data are presented as mean values ± SD (n = 7–8, biological replicates). P values were calculated by two-way ANOVA with post hoc Bonferroni test; not significant (ns). Source data: Source Data file (6c, d).

causes a decrease in the half-life of these transcripts by several times and is sufficient for significant reduction of Ig levels in the mice leading to a deficient response to immunization with TI antigen. This proves that cytoplasmic polyadenylation is not restricted to deadenylated maternal mRNAs, as in the case of gametogenesis. However, the mechanism of the substrate specificity and regulation of TENT5C-mediated Ig polyadenylation remains to be established.

*Tent5c* was reported as one of the genetic signatures in ASC and a potential regulator of B cell differentiation, and is one of the top 50 upregulated genes in the spleen and BM PCs (ref. [42]). Our studies confirm a strong correlation of TENT5C expression with B cell proliferation and differentiation into PCs. Moreover, it is positively correlated with the upregulation of PABPC1 previously identified as TENT5C interactor in MM cells[21]. Innate signaling from both surface (TLR1, 2, 4, and 6) and intracellular (TLR9) TLRs strongly upregulate TENT5C levels, thus promote the B cell lineage differentiation and enhance immune response; however, detailed dissection of TLR downstream signaling pathways require further investigation[43]. Interestingly, mice devoid of MyD88 (myeloid differentiation primary response gene 88) gene, which is one of the key elements of TLR signaling, reveal similar phenotypes to TENT5C deletion including: decreased steady-state levels of total serum Igs, decreased antigen-specific IgM and IgG1 antibody responses, and abolished IgG2 antibody response in immunized mice[44]. This strongly suggests that TENT5C is one of

the TLR signaling effectors in B cells. In agreement, TENT5C upregulation by specific B cell innate signaling is underlined by the fact that it is not affected by the stimulation of BCR and CD40 receptor, and also that a decrease in humoral response was only observed in mice immunized with T-cell-independent antigens.

The transition of naive B cells into ASC requires significant ER membrane expansion, given that the structural maturation of Igs is facilitated by ER-residing chaperones and folding machinery[1,45–47]. The increase in the volume of secretory organelles occurs through the generation of ER sheets and requires UPR signaling[48]. It has been shown that XBP1 and BLIMP1 transcription factors are required for PC development and they link B cell physiology with the UPR response[49–51]. Accordingly, the analysis of ASC signature genes identified 30% of the transcriptome as related to the UPR (ref. [42]). Thus, the downregulation of Ig expression in *Tent5c* KO has to impair the UPR, which plays a pivotal role in the differentiation of ASC. In agreement with this, basic ER stress in naive B cells from TENT5C-deficient mice is enhanced, which reflects their faster proliferation and differentiation into PC, while ER expansion dynamics is reduced. This also shows that cytoplasmic adenylation by TENT5C, being the posttranscriptional regulator of Ig expression, may have profound effects on different aspects of B cell physiology. Although TENT5C activity is predominantly cytoplasmic, it should be marked that further studies are required to exclude its effect on nuclear mRNA tailing.

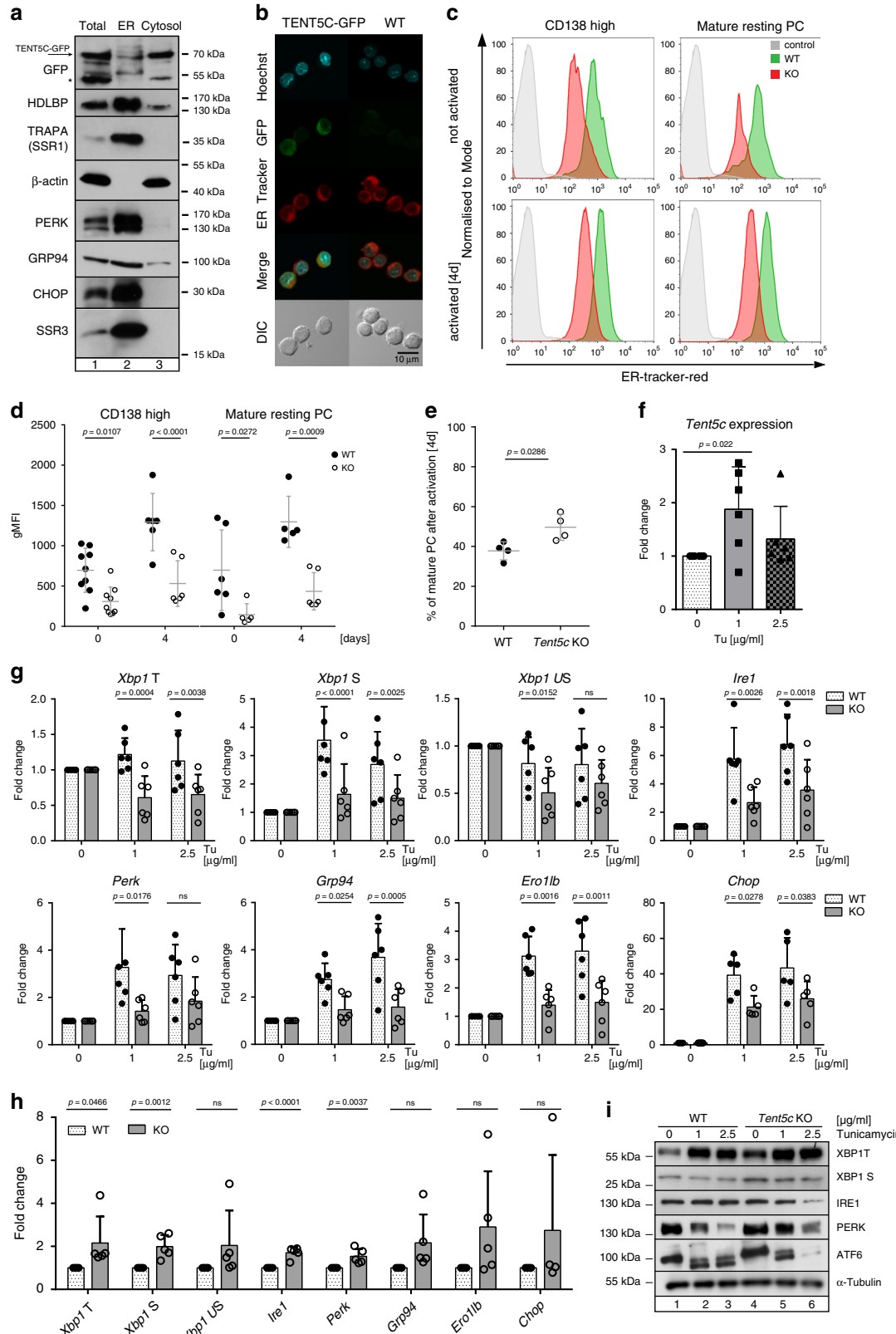

In conclusion, this study identified ncPAP TENT5C as a new factor involved in the regulation of Ig production through ER-associated polyadenylation in B cell lineage in mice. Thus, we demonstrate the significance of cytoplasmic poly(A) tail homeostasis as an important regulatory level for B cell immune response.

## Methods

**Mice**. Mice strains used in this study are listed in Supplementary Table 2. Mice strains: B6CBAF1;B6-TENT5C KO/Tar; B6CBAF1;B6-TENT5C Cat (D90N; D92N)/Tar; and B6CBAF1;B6-TENT5C-GFP/GFP/Tar were generated and geno-typed by the Mouse Genome Engineering Facility (www.crisprmice.eu). Experimental mice originated from heterozygotic matings and they were cohoused

**Fig. 7 TENT5C affects the UPR response. a** TENT5C is present in the ER fraction. Western blot analysis of *Tent5c*-GFP B cells (activated for 7 days), fractionation for ER, and cytosol. Arrow indicates the position of TENT5C-GFP, while asterisks indicate nonspecific bands. β-actin, HDLBP, SSR1, SSR3, PERK, CHOP, and GRP94 were used as fractionation controls. **b** Staining of the nucleus and the ER of activated B cells isolated from WT and *Tent5c*-GFP mice. **c** Flow cytometry analysis of ER-fluorescence intensity in CD138[high] cells and mature resting PC from WT and *Tent5c* KO at day 0, and 4 days after activation. PC populations were analyzed based on CD138, CD19, CD45R, and ER-tracker. See the gating strategy in Supplementary Fig. 4. **d** Quantification of ER volume, measured as gMFI (geometric mean of fluorescence intensity) of ER-tracker, based on flow cytometry. gMFI for ER-tracker was checked for CD138[high] cells (left panel) and mature resting PC (right panel) at day 0, and 4 days after activation ($n = 5$–9). **e** Percentage of mature PC 4 days after activation ($n = 4$). **f** qPCR analysis of *Tent5c* expression level after tunicamycin (Tu) treatment. WT B cells activated for 3 days, then treated with Tu in indicated concentrations for 5 h ($n = 6$). **g** qPCR analysis of UPR markers expression in WT and *Tent5c* KO cells after ER stress induction with Tu in third day of activation (*Xbp1* S—spliced, US—unspliced, T—total; $n = 6$ or *Chop* $n = 5$). **h** qPCR analysis of basal ER stress level in B cells WT and *Tent5c* KO in third day after activation ($n = 5$, *Chop* $n = 4$). **i** Western blot analysis of UPR markers in WT and *Tent5c* KO cells after ER stress induction with Tu. The α-tubulin was used as a loading control. $P$ values were calculated using two-way ANOVA with post hoc Bonferroni test **d**, **g**, two-tailed Mann–Whitney $U$ test **e**, two-tailed unpaired Student's $t$-test in **f**, **h**; not significant (ns); data are presented as mean values ± SD; $n$—biological replicates. Source data: Supplementary Fig. 8 (7a, i) and Source Data file (7d–h).

littermates. Mice of both sex were sacrificed at age 12–16 weeks (*Tent5c* KO and *Tent5c*-GFP) and 48–56 weeks (*Tent5c* Cat). All mice were bred in the animal house of Faculty of Biology, University of Warsaw. The immunization experiment was held at the animal house of Medical University of Warsaw. Mice were maintained in conventional conditions in open polypropylene cages filled with wood chip bedding (Rettenmaier). Environment was enriched with nest material and paper tubes. Mice were fed at libutum with standard laboratory diet (Labofeed B, Morawski). Humidity in the rooms was kept at 55 ± 10%, temperature at 22 °C ± 2 °C, at least 15 air changes per hour, and light regime set at 12 h/12 h (lights on from 6:00 to 18:00). Health monitoring was performed regularly at the IDEXX laboratory and reports are shown in Supplementary Table 3. The mice used in this study were naive and had no previous history of experimentation or exposure to drugs. The *Tent5c* KO mice strain was described previously[21]. The *Tent5c* C-terminal-GFP knock-in and *Tent5c* Cat animals were generated, and genotyped by the Mouse Genome Engineering Facility. All procedures were approved by the I Local Ethical Committee in Warsaw affiliated at Univeristy of Warsaw, Faculty of Biology (approval number WAW/176/2016) and II Local Ethical Committee in Warsaw, affiliated at Warsaw University of Life Sciences (approval number WAW2/159/2019), with the requirements of the EU (Directive 2010/63/EU) and Polish (Act number 266/15.01.2015) legislation.

**Generation of *Tent5c*-GFP knock-in and *Tent5c* Cat mice.** The *Tent5c* C-terminal-GFP knock-in mice line was generated using single-guide RNA (chimeric sgRNA). The sgRNA was synthesized using T7 RNA polymerase and DNA template obtained with mTENT5C_GFP_sgRNA_F and Universal_gRNA_rev primers (T7 promoter is underlined). All subsequent steps were performed as described previously[21]. The dsDNA donor for in-frame C-terminal TEV-GFP knock-in, two 1 kb homology arms flanking *Tent5c* STOP codon were amplified from C57BL/6 J gDNA using mTent5C_TOPO-LF_1f/mTent5C_LF-TEV_1r and mTent5C_eGFP-RF_1f/mTent5C_RF-TOPO_1r primer pairs. The TEV-eGFP coding sequence was amplified from pKK-TEV-eGFP plasmid using TEV_1F and mCherry_GFP_1R primers. Assembly of TEV-eGFP and 1 kb homology arms flanking *Tent5c* STOP codon was performed with TOPO Zero Blunt (Thermo Fisher Scientific). Next, we amplified 865 bp fragment of TENT5C-TEV-eGFP donor flanked with 60 bp homology arms from pTOPO/F46C/TEV-eGFP and mTent5C-GFP_short_1F/mTent5C-GFP-short_1R primers. PCR product was purified with Agencourt AmpureXP magnetic beads (Beckman). Mice genotyping was carried out in PCR reaction using mTent5C_GFP_seqF and mTent5C_GFP_seqR primers, and Phusion HotStart II Polymerase (Thermo) and gDNA isolated from of mice ears or tails fragments with HotShot method or with Genomic Mini DNA isolation kit (A&A Biotechnology). *Tent5c* catalytic mutants (*Tent5c* Cat), harboring two missense mutations D90N and D92N were created using the CRISPR/Cas9 method. sgRNA and SpCas9 mRNA sequences were identical to those used in creating *Tent5c* KO and were described previously[21]. For introducing two missense mutations 85 nt TENT5C_catODN oligonucleotide was added to the microinjection mix at the final concentration of 1.25 pM. *Tent5c* Cat mice were genotyped by Sanger sequencing of 585 bp amplicons obtained with PCR using Fam46C_seq2F and Fam46C_seq2R primers. Chromatograms were analyzed using Mutation Surveyor 4.0 (SoftGenetics). All primers used for mice generation are listed in Supplementary Table 4.

**Immunization of mice.** To study the thymus-dependent (TD) antibody responses, 12-week-old *Tent5c* KO and WT littermates were immunized intraperitoneally with chicken OVA (Sigma-Aldrich) antigen (50 µg/mouse) mixed with complete Freud adjuvant (CFA, Invivogen; 1:1 vol:vol) at days 0 and 14. Mice were bled on indicated days via facial vein puncture. The concentrations of anti-OVA Igs were measured with the ELISA kit (Cayman Chemical, 500830) according to the manufacturer's recommendation. For TI antibody responses, 12-week-old *Tent5c* KO and WT littermates were immunized with TNP-LPS (50 µg/mouse, Bioresearch

Technologies, T-5065-5). Mice serum was collected at days 0, 7, and 14. The concentrations of anti-TNP Igs were measured with the ELISA kit (Life Diagnostics, Inc) according to the manufacturer's recommendation. The experiments were performed in accordance with the guidelines approved by the II Local Ethics Committee in Warsaw (approval no. WAW2/159/2019) and in accordance with the requirements of the EU (Directive 2010/63/EU) and Polish (Dz. U. poz. 266/15.01.2015) legislation.

**Serum Igs ELISAs.** Total concentrations of Igs of different isotypes in the sera collected from WT and *Tent5c* KO mice were quantified using Mouse Ig Isotyping Ready-SET-Go (Invitrogen, 88-50630) and standards: IgA (Invitrogen, 39-50450-65), IgG1 (Invitrogen, 39-50410-65), IgG2a (Invitrogen, 39-50420-65), IgG2b (Invitrogen, 39-50430-65), IgG3 (Invitrogen, 39-50440-65), and IgM (Invitrogen, 39-50470-65) according to the manufacturer's recommendation.

**Tissue collection and blood analysis.** Blood samples were collected terminally from the mandibular vein to EDTA or serum separator tubes. Complete blood counts, gel electrophoresis of proteins, and serum iron level analysis were performed at the Veterinary Diagnostic Laboratory LabWet in Warsaw (http://www.labwet.pl/) on the day of blood collection. SPEP analyses were performed using SAS-MX SP-10 Kit (Helena-Biosciences). Serum samples were diluted in the buffer in a ratio of 1:4 and proteins were separated at a constant voltage of 80 V through 25 min. Gels were quantified with Platinum software V6 (Helena-Biosciences). All mice were sacrificed by cervical dislocation. Spleen and femur and tibia bones were isolated immediately. BM was isolated using centrifugation method[52]. BM was depleted of red blood cells using ACK lysis buffer (154.95 mM ammonium chloride, 10 mM potassium bicarbonate, and 0.1 mM EDTA).

**Primary cell culture and ex vivo B cell activation.** A single-cell suspension of splenocytes was obtained by mechanical tissue disintegration of the spleen through a 70 µm cell strainer. Then, splenocytes were additionally depleted from red blood cells using ACK lysis buffer before separation. Naive B cells were isolated from spleen using immunomagnetic negative selection with EasySep™ Mouse B Cell Isolation Kit (Stemcell; 19854) and CD138[high] cells were isolated from spleen and BM, with EasySep Mouse CD138 Positive Selection Kit (Stemcell; 18957) according to the manufacturer's instructions. Primary cells were cultured in RPMI 1640 ATCC's modified (Invitrogen) supplemented with 15% FBS (Invitrogen), 100 nM 2-mercaptoethanol (Sigma), penicillin/streptomycin (Sigma), and activators or mitogens depending on the experiment: 20 ng/ml IL-4 (Peprotech; 214-14), 0.5 µg/ml megaCD40L (Enzo Lifesciences; ALX-522-12-C010), 10 µg/ml anti-IgM (Thermo Fisher Scientific 16-5092-85), 20 µg/ml LPS (Santa Cruz Biotechnology; sc-3535), 1 µg/ml Pam3CSK4 (Invivogen; tlrl-pms), $10^8$ cells/ml HKLM (Invivogen, tlrl-hklm), 10 µg/ml Poly(I:C), HMW and LMW (Invivogen; tlrl-pic and tlrl-picw), 10 µg/ml LPS-EK standard (Invivogen; tlrl-peklps), 1 µg/ml FLA-ST (Invivogen; tlrl-epstfla), 100 ng/ml FSL1 (Invivogen; tlrl-fsl), 1 µg/ml ssRNA40/LyoVec (Invivogen; tlrl-lrna40), and 1 µM ODN1826 (Invivogen; tlrl-1826). In most experiments, cells were activated with 20 ng/ml IL-4 and 20 µg/ml LPS, unless stated otherwise.

**Spleen histology and PC microscopy.** For the histology of the spleen, animals were overdosed with ketamine/xylazine and perfused transcardially at 10 ml/min flow rate with PBS for 1 min following 4% PFA in phosphate buffer (PB) for 2 min at room temperature (RT). Organs were dissected, postfixed in PFA for 2 h at RT and suffused with 30% sucrose solution in PB overnight at 4 °C. For immuno-histochemical staining, 10-µm thick sections were cut with the cryostat, and sections were blocked 2 h with 10% rabbit serum and 1% BSA in TBS with 0.3% Triton X-100. Sections were incubated with primary antibodies anti-GFP from chicken (Abcam, ab13970) and anti-CD138 PE from rat (Clone 281-2) diluted

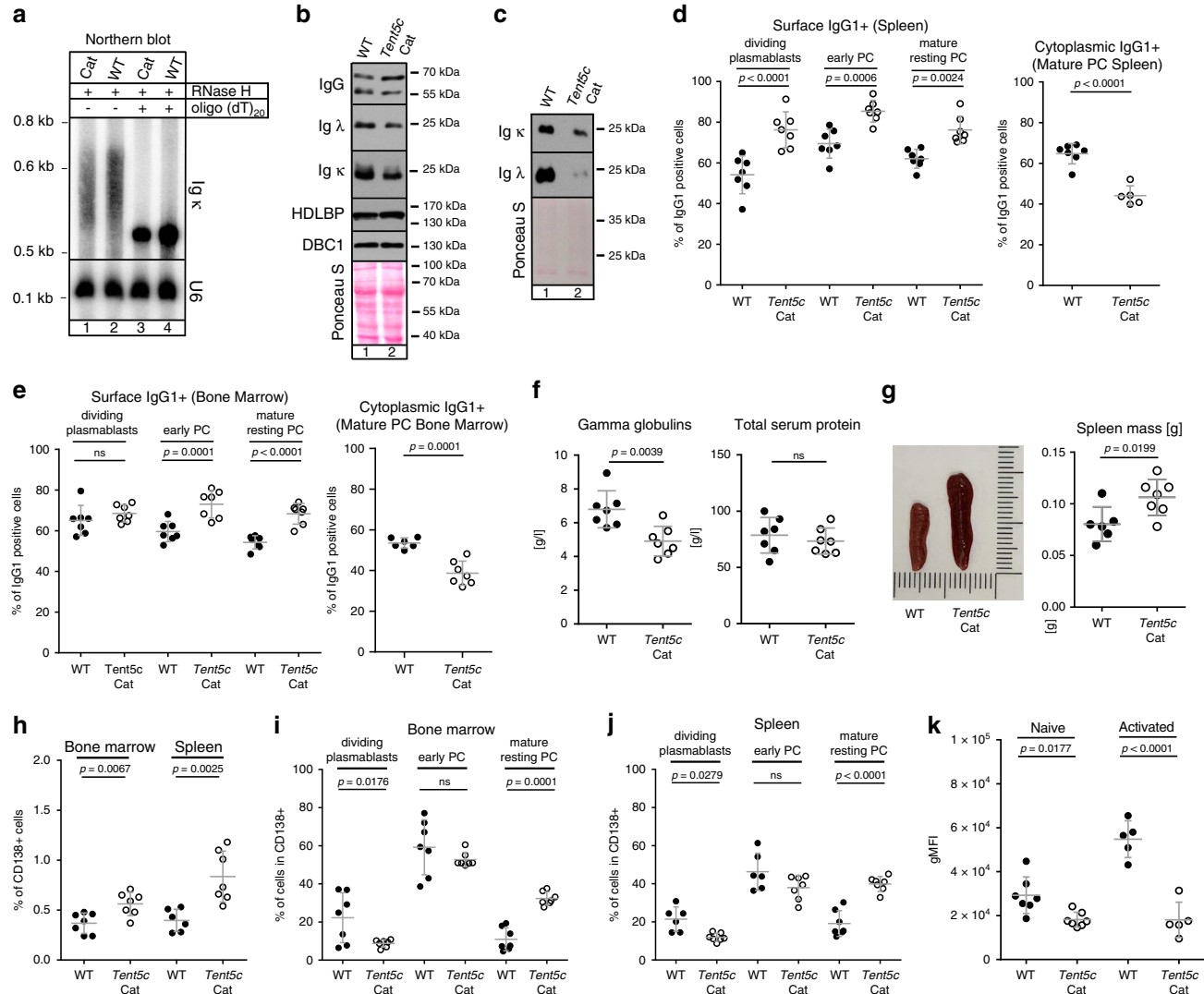

**Fig. 8 Loss of TENT5C catalytic activity leads to humoral immune response deficiency. a** The Ig κ mRNA poly(A) tail is reduced in *Tent5c* Cat B cells. Northern blot analysis of Ig κ mRNA and U6 RNA (loading control) from *Tent5c* Cat and WT B cells activated for 7 days, treated with RNase H in the presence of oligo(dT)$_{20}$. **b, c** Western blot analysis of extracellular IgG heavy and light (κ and λ) chains, HDLBP, and DBC1 **b** or secreted to media Igs κ and λ light chains **c** from WT and *Tent5c* Cat B cells activated for 3 days. Ponceau S staining was used as a loading control. **d, e** Percentage of cells positive for surface and cytoplasmic IgG1 isolated from spleen **d** or BM **e** (**d** $n = 7$ or $n = 5$ for Cat in cytoplasmic IgG1+; **e** $n = 7$ or $n = 6$ for WT in cytoplasmic IgG1+). **f** Examination of blood serum gamma globulins and total protein levels in *Tent5c* Cat and WT by SPEP ($n = 7$). **g** Comparison of spleen mass in WT and *Tent5c* Cat mice (right panel; WT $n = 6$, Cat $n = 7$). Representative pictures are shown in the left panel. **h** Percentage of CD138-positive cells in BM and spleen; ($n = 7$ or $n = 6$ for WT in spleen). **i, j** Comparison of CD138-positive subpopulations: dividing plasmablasts, early PC, mature resting PC in WT, and *Tent5c* Cat isolated from BM **i** and spleen **j** (**i** $n = 7$; **j** $n = 7$ or $n = 6$ for WT dividing plasmablasts and early PC). **k** Quantification of ER volume, measured as gMFI (geometric mean of fluorescence intensity) of ER-tracker stained mature resting PC at day 0 (naive, $n = 7$) and 4 days after activation (activated, $n = 5$). $P$ values were calculated using two-way ANOVA with post hoc Bonferroni test **d, e, i–k**, two-tailed unpaired Student's $t$-test **d, e** left panels, **f, h**, two-tailed unpaired Student's $t$-test with Welch correction **g**; not significant (ns). Data are presented as mean values ± SD; $n$—biological replicates. Source data: Supplementary Fig. 8 (8a–c, g) and Source Data file (8d–f, h–k).

---

1000× and 500×, respectively, in blocking solution and developed with goat anti-Chicken Alexa 488 (Thermo Fisher Scientific, A-11039) with Hoechst diluted in blocking solution. After quenching and incubation with antibodies, sections were washed three times 5 min with TBS with Triton X-100 0.025%.

Isolated CD138$^{high}$ cells were stained using ER Staining Kit (Abcam; ab139482) according to the manufacturer's instruction with the following exceptions: Red Detection Reagent was diluted 2000×, Hoechst 33342 Nuclear Stain was diluted 500×, and the staining time was shortened to 4 min.

PCs and spleen section imaging were performed using a confocal system (Fluoview FV1000) equipped with a spectral detector (Olympus) and with a 60× oil objective with 1.40 aperture. Images were processed using ImageJ 1.52p software.

**GST-eIF4E$^{K119A}$ purification.** Chemocompetent *E. coli* BL21-CodonPlus-RIL strain (Stratagene) was transformed with plasmid pGEX-4T-3 carrying GST-eIF4E$^{K119A}$.

Cells were preincubated in standard Luria-Broth (LB) medium (with 0.1 mg/ml ampicillin and 34 mg/ml chloramphenicol) overnight and then transferred to Auto Induction Media Super Broth Base Including Trace Elements (Formedium) supplemented with 2% glycerol, kanamycin (50 mg/ml), and chloramphenicol (34 mg/ml) and incubated for 48 h in 18 °C with shaking 150 rpm. Bacteria were pelleted by centrifugation at 4500 rpm for 15 min at 4 °C, frozen in liquid nitrogen and stored at −20 °C. Pellet from 4 l of culture was used for single purification. Pellet was resuspended in column buffer (50 mM $N_2HPO_4$ pH 7.5, 150 mM NaCl, 1 mM EDTA) supplemented with 1 mM DTT, 1% Triton X-100, 1 mM PMSF, a protein inhibitor cocktail (20 nM pepstatin, 6 nM leupeptin, 2 ng/ml chymostatin), and 50 μg/ml lysozyme, incubated for 20 min in 4 °C, then broken in a French pressure cell press MultisiFlex-C3 at 500 Bar. The homogenate was centrifuged in a Sorvall WX ULTRA SERIES ultracentrifuge, F37L rotor at 32000 rpm for 45 min at 4 °C. The supernatant was loaded on a 1 ml column with Glutathione Sepharose 4B resin (GE Healthcare; 17-0756-01), equilibrated by column buffer. ÄKTA Purifier system

(GE Healthcare) was used for all purification steps. Unbound proteins were washed out with 20 CV of column buffer. GST-eIF4E was collected during 5 CV washing of elution buffer (50 mM Na$_2$HPO$_4$ pH 8.5, 10 mM L-glutathione reduced, 1 mM DTT). The elution fraction was mixed and diluted three times with water and loaded into the ion-exchange column (Resource S GE Healthcare; 17-1180-01), equilibrated by 50 mM Na$_2$HPO$_4$ pH=8, 100 mM NaCl. GST-eIF4E was eluted by 0.1–1 M NaCl gradient. All purification steps were analyzed by SDS–PAGE.

**RNA isolation**. Total RNA was isolated from cells with TRIzol reagent (Thermo Fisher Scientific) according to the manufacturer's instructions, dissolved in nuclease-free water and stored at −20 °C.

**RNase H assay**. Total RNA (20 µg) isolated from activated B cells, was mixed with 100 ng of IgK_RNaseH oligo (which hybridizes to Ig κ sequence and allow to shorten this transcript; sequence provided in Supplementary Table 5) and with (or without as a control) 100 ng of oligo(dT)$_{20}$, heated for 1 min at 95 °C, and cooled down to 37 °C. Then, 10 µl of 5× RH Reaction buffer (NEB) and 5 U of RNase H (NEB) were added. Reactions were carried out for 45 min at 37 °C and RNA were recovered by extraction with phenol/chloroform, precipitated, and analyzed using northern blots.

**RT-qPCR**. For the quantitative analysis, RNA was first treated with DNase (TURBO DNA-free Kit, Invitrogen; AM1907) for 30 min at 37 °C and then reverse transcribed using SuperScript III (Invitrogen; 18080085) and oligo(dT)$_{20}$ and random-primers (Thermo Fisher Scientific). The quantitative PCR was performed with Platinum SYBR Green qPCR SuperMix-UDG (Thermo Fisher Scientific; 11733046) using LightCycler 480 II (Roche) PCR device and appropriate primers listed in Supplementary Table 5. Gene expression for each sample was normalized to GAPDH. Differences were determined using the $2^{-\Delta\Delta C(t)}$ calculation.

**Northern blotting**. RNA samples were separated on 4% acrylamide gels containing 7 M urea in 0.5× TBE buffer and transferred to a Hybond N+ membrane by electrotransfer in 0.5× TBE buffer. After transfer, membranes were stained with 0.03% methylene blue in 0.3 M NaAc pH 5.3 for 5 min at RT, scanned, and then destained with water. RNA was immobilized on membranes by 254 nm UV light using a UVP CL-1000 crosslinker. Radioactive probes were labeled with an α$^{32}$P (dATP) with a DECAprime II DNA Labeling Kit (Thermo Fisher Scientific; AM1455). To obtain templates for probes labeling, PCR on cDNA from B cells activated with LPS and IL-4 for 7 days was conducted using primers listed in Supplementary Table 5. Alternatively, an oligo probe for U6 as a loading control was labeled with a γ$^{32}$P (ATP) with T4 PNK (NEB).

Membranes were pre-hybridized in PerfectHyb Plus Hybridization Buffer (Sigma, H7033) for 1 h 65 °C and incubated with radioactive probes in PerfectHyb Plus Hybridization Buffer overnight 65 °C (in case of oligo probe—at 37.5 °C). Then membranes were washed in 2× SSC with 0.1% SDS for 20 min, 0.5× SSC with 0.1% SDS for 20 min and 0.1× SSC with 0.1% SDS for 20 min, scanned with Fuji Typhoon FLA 7000 (GE Healthcare Life Sciences), and analyzed with Multi Gauge software V3.0 (Fujifilm).

**mRNA enrichment with GST-eIF4E$^{K119A}$ protein**. Purification was performed as described previously with some modifications[25]. Glutathione Sepharose 4B resin was incubated with GST-eIF4E$^{K119A}$ protein in sterile PBS (200 µl resin per 200 µg protein) for 1 h at RT with rotation. Then, the resin was washed two times with PBS and three times with buffer B (10 mM potassium PB, pH 8.0, 100 mM KCl, 2 mM EDTA, 5% glycerol (Sigma), 0.005% Triton X-100 (Sigma), 6 mM DTT (A&A Biotechnology), and 20 U/mL Ribolock RNase Inhibitor (Thermo Fisher Scientific, EO0381). A total of 100 µg of total RNA, previously denaturated 10 min at 70 °C, was mixed with the prepared resin and incubated for 1 h at RT on an immunoprecipitation rotor. Then, the resin was washed three times with buffer B, two times with buffer B supplemented with 0.5 mM GDP (Sigma), and two times with buffer B without GDP. RNA was eluted from the resin by acid phenol: chloroform extraction and precipitated using 100% ethanol (Merck), 3 M sodium acetate, and GlycoBlue co-precipitant (Thermo Fisher Scientific).

**Validation of the mRNA enrichment with GST-eIF4E$^{K119A}$ protein**. Total RNA (150 µg) from HEK293T cells (which was a gift from Dr. Agnieszka Tudek) was subjected to mRNA enrichment with GST-eIF4E$^{K119A}$ protein as described above. Then, both total and purified RNA were either used for northern blot analysis or treated with DNase and 300 ng was used for reverse transcription (as described above). To estimate mRNA enrichment and rRNA-removal efficiency cDNA was used for qPCR analysis, as described above.

**RNA-seq**. All experiments were done in triplicate, where a single replicate originated from *Tent5c* KO and WT littermates at age 12–15 weeks.

*Cell culture and RNA retrieval*: Naïve B cells were isolated and cultured as described above. Subsequently, after isolation cells were activated by the addition of 20 ng/ml IL-4 (Peprotech) and 20 µg/ml LPS (Santa Cruz) to the medium, followed by 7 days of incubation. Finally, RNA was isolated as described above.

*Library preparation*: Total RNA was treated with DNase for 30 min in 37 °C and 1 µg of RNA was subjected to ribodepletion using a Ribo-Zero Gold rRNA-removal kit H/R/M (Illumina), according to the manufacturer's recommendations and spiked-in with external RNA (ERCC RNA Spike-In Mix, Thermo Fisher Scientific; 4456740). Strand-specific libraries were prepared using a dUTP protocol (KAPA Stranded RNA-Seq Library Preparation Kit, KAPA Biosystems; KK8401), according to the manufacturer. Library quality was assessed using chip electrophoresis performed on an Agilent 2100 Bioanalyzer (Agilent Technologies, Inc.). The libraries were sequenced using an Illumina NextSeq500 sequencing platform to an average number of ~$1.5 \times 10^7$ reads per library in the 75-nt paired-end mode.

**Nanopore direct RNA sequencing and polyadenylation analysis**. *RNA retrieval*: Direct RNA sequencing was performed in triplicate. For two replicates, the same input RNA (batches 1 and 2) as for the RNA-seq experiment were used (described above). RNA (100 µg) was subjected to mRNA enrichment with GST-eIF4E$^{K119A}$ protein (as described above), followed by the ribodepletion of 2.5 µg RNA using a Ribo-Zero Kit (Illumina), according to the manufacturer's recommendations. The remaining replicates (batch 3) were prepared without ribodepletion step.

*Library preparation and sequencing*: Nanopore direct RNA sequencing libraries were prepared from 500 ng of cap-enriched and ribodepleted mRNA with Direct RNA Sequencing Kit (ONT, SQK-RNA001 (for replicates 1–2) or SQK-RNA002 (replicate 3)). In the case of replicate 3, 4 µg cap-enriched mRNA (without ribodepletion) was mixed with the 150 ng of *Saccharomyces cerevisiae* oligo(dT)-enriched RNA and 50 ng of in vitro transcribed poly(A) standards. All remaining steps were performed according to the manufacturer's instructions, including the reverse transcription step. Sequencing was performed with the MinION device, MinKNOW 19.10.1 software, Flow Cell (Type R9.4.1; RevC (RevD for replicate 3)), and basecalled using Guppy 3.3.0 (ONT).

**Actinomycin D treatment**. B cells activated for 3 days were treated with actino-mycin D (Sigma) at a final concentration of 5 µg/ml for 0–4 h. Then, RNA was isolated with TRI reagent and subjected to RT-qPCR analysis as described above. Half-lives were estimated by nonlinear regression (one-phase decay analysis) in GraphPad Prism 6.

**4-thiouridine labeling**. B cells from WT and *Tent5c* KO mice were activated with LPS (20 µg/ml) and IL-4 (20 ng/ml) for 7 days. Then cells were treated with 4sU (Sigma, T4509) in a final concentration of 50 µM for 1 h and cells were collected. RNA was isolated using TRI reagent and treated with TURBO DNase as described above. To minimize sample-to-sample variation, a previously prepared spike-in (RNA from *Schizosaccharomyces pombe* labeled with 4-thiouracil) was added to each sample. Then, RNA was biotinylated using MTSEA-biotin-XX (Biotium, 90066). The reaction containing 125 µl of RNA, 10 µl of 10× biotinylation buffer (100 mM HEPES pH 7.5, 10 mM EDTA), and 10 µl of 1 mg/ml MTSEA-biotin-XX was incubated in dark for 30 min on a rotating wheel. Then RNA was purified using Phase Lock Gel heavy tubes (5PRIME) and precipitated using 100% ethanol, 3 M sodium acetate, and 1 µl GlycoBlue (Invitrogen). Biotinylated RNA was isolated with Dynabeads MyOne Streptavidin C1 beads (Invitrogen). To 240 µl RNA, 30 µl of 10× NaMg (100 mM Tris-HCl pH 7.0, 2 M NaCl, 250 mM MgCl$_2$), 30 µl of 1 M NaPi buffer pH 6.8, and 3 µl of 10% SDS were added. For one isolation, the 50 µl of beads were used. Beads were washed with 400 µl NaMgPS buffer (10 mM Tris-HCl, pH 7.0, 200 mM NaCl, 25 mM MgCl$_2$, 100 mM NaPi pH 6.8, 0.1% SDS) once and blocked with 10 µl of 20 mg/ml glycogen (Invitrogen) in 200 µl NaMgPS for 20 min. Beads were washed with 400 µl NaMgPS buffer and then incubated with RNA for 30 min in dark on a rotating wheel. Beads were washed five times with 400 µl NaMgPS and once with TEN1000 (10 mM Tris-HCl, pH 7.5, 0,5 mM EDTA, 1 M NaCl). To elute RNA, beads were suspended in 100 µl of 100 mM DTT, incubated for 5 min, placed on magnetic stand, and supernatant was collected as nascent RNA fraction. The elution was repeated once. RNA was precipitated using 100% ethanol, 3 M sodium acetate, pH 5.3, and 1 µl GlycoBlue (Invitrogen). Then nascent and total RNA fractions were subjected to RT-qPCR analysis as described above. Half-lives were estimated using equation: $t_{1/2} = -t_L \times \ln(2)/\ln(1-nascent/total)$[53].

**Co-IP and MS**. Activated with LPS (20 µg/ml) and IL-4 (20 ng/ml) *Tent5c*-GFP B cells were crosslinked with 1 mM DSP (dithiobis(succinimidyl propionate); Invitrogen) for 1 h before stopping the reaction with 50 mM Tris pH 8.0. After washing with PBS-supplemented 50 mM Tris pH 8.0, cells were flash-frozen in liquid nitrogen, thawed on ice, and incubated for 30 min at 4 °C with gentle rotation in 3 ml LB buffer (50 mM Tris, 150 mM NaCl, 0.5% Triton X-100, 1 mM DTT, supplemented with proteases, and phosphatase inhibitors; Invitrogen). Next, the lysates were sonicated for 30 min with a Bioruptor Plus (Diagenode), followed by clarification by centrifugation. Immunoprecipitations were performed using a GFP-Trap (ChromoTek; gtm-100). After 2 h of incubation, the beads were washed six times with LB buffer and finally, the proteins were eluted with 50 mM glycine pH 2.8. After neutralization with Tris pH 8.0, proteins were precipitated with PRM reagent (0.05 mM pyrogallol red, 0.16 mM sodium molybdate, 1 mM sodium

oxalate, 50 mM succinic acid; pH 2.5 (Sigma-Aldrich)) prior to MS analysis in the Laboratory of MS, IBB PAS.

**Western blotting**. For western blot analysis, equal amount of cells were lysed with 0.1% NP40 in PBS supplemented with protease inhibitors and viscolase (A&A Biotechnology, 1010-100) for 30 min in 37 °C with shaking 600 rpm, then Laemmli buffer was added and samples were denatured for 10 min in 100 °C. For analysis of secreted proteins, the cell culture medium was collected and centrifuged twice 15 min 13,500 rpm, then the supernatant was collected, Laemmli buffer was added, and samples were denatured for 10 min at 100 °C. Samples were separated on 12–15% SDS–PAGE gels, proteins were transferred to Protran nitrocellulose membranes (GE Healthcare), and then membranes were stained with 0.3% w/v Ponceau S in 3% v/v acetic acid and digitized. Membranes were incubated with 5% milk or 5% BSA in TBST buffer according to the technical recommendations of the antibodies' suppliers for 1 h followed by incubation with specific primary antibodies (catalog and lot numbers are listed in the Supplementary Table 6 and Reporting Summary file) diluted 1:25,000 (β-actin, Clone C4), 1:10,000 (α-tubulin, Clone DM1A), 1:5000 (HDLBP Bethyl, A303-971A; IgG Millipore, 401215; Igλ SCBT, sc-516132; Igκ SCBT, sc-516102), 1:3000 (IL-6, Clone 10E5; GRP94, Clone H-212), 1:2000 (PERK, Clone C33E10; Xbp1, Clone EPR22004; TRAPα, Clone EPR5603; SSR3 Abcam, ab190936), or 1:1000 (Ire1, Clone 14C10; ATF6, Clone D4Z8V; CHOP, Clone D46F1; GFP ChromoTek, PABG1-100) overnight in 4 °C. Membranes were washed three times in TBST buffer, incubated with HRP-conjugated secondary antibodies: anti-mouse (Millipore, 401215) diluted 1:5000 and anti-rabbit (Millipore, 401393) diluted 1:3000, for 2 h at RT. Membranes were washed three times in TBST buffer and proteins were visualized by enhanced chemiluminescence acquired on X-ray film.

**Flow cytometry**. Splenocytes and BM were isolated as described above, and after depletion of red blood cells with ACK buffer cells were stained respectively. Designed staining panels were based on the "Flow cytometry tools for the study of B cell biology" (BD Pharmingen). The antibodies used for flow cytometry analysis are listed in Supplementary Table 6. Samples were measured with BD LSRFortessa™ under FACS Diva Software v8.0.1 (BD) software control and analyzed using FlowJo (Data Analysis Software v10).

**Surface staining of early developmental B cell stages**. Cells ($1.5 \times 10^6$) isolated from the BM or spleen were pelleted and incubated with Fc block (anti-CD16/32) for 10 min at RT. After washing with FACS buffer (0.2% BSA in PBS), cells isolated from BM were stained with anti-CD19 BUV395 (Clone 1D3), anti-CD43 BV421 (Clone S7), anti-CD23 BV421 (Clone B3B4), anti-IgM FITC (Clone II/41), anti-CD249 PE (Clone BP-1), anti-CD93 PE (Clone AA4.1), anti-IgD BV605 (Clone 11-26 c.2a), anti-CD45R/B220 APC (Clone RA3-6B2), anti-CD24 PerCP Cy5.5 (Clone M1/69), and anti-CD21/CD35 PerCP Cy5.5 (Clone 7G6) antibodies for 30 min in 4 °C, protected from light. In case, when two antibodies produced with BD Horizon technology were used in one staining panel, Brilliant Stain Buffer (BD Biosciences; 566349) was used to prepare the antibody mix. After staining, cells were washed with FACS buffer and then stained with LIVE/DEAD™ Fixable Near-IR Dead Cell Stain Kit (Thermo Fisher Scientific; L34976) for 20 min in 4 °C, protected from light. Finally, cells were washed and analyzed in terms of calculation PreProB, ProB, PreB, immature, transitional B, early and late mature B, T1, T2, T3, follicular B I and II, and marginal cell subsets (Marginal Zone and Marginal Zone Progenitor).

**Surface staining of plasmablasts and PCs**. Cells ($2.5 \times 10^6$) isolated from the BM or spleen were incubated with Fc block anti-CD16/32, (Clone 2.4G2) for 10 min at RT, washed with FACS buffer and incubated with anti-CD19 BUV395 (Clone 1D3), anti-CD19 AF700 (Clone 1D3), anti-IgM FITC (Clone II/41), anti-IgA PE (Clone 11-44-2), anti-CD138 BV605 (Clone 281-2), anti-CD138 PE (Clone 281-2), anti-IgG1 AF647 (Thermo Fisher Scientific A-21240), and anti-CD45R/B220 PerCP Cy5.5 (Clone RA3-6B2) antibodies for 30 min in 4 °C, protected from light. In case, when two antibodies produced with BD Horizon technology were used in one staining panel, Brilliant Stain Buffer was used to prepare the antibody mix. Next, cells were washed with FACS buffer and then stained with LIVE/DEAD™ Fixable Near-IR or Violet Dead Cell Stain Kit (Thermo Fisher Scientific; L34964 for 20 min in 4 °C, protected from light. Finally, cells were washed and analyzed in terms of calculation dividing plasmablasts, early plasmocytes, and mature resting plasmocytes.

**Staining of intracellular Igs**. Cells ($2.5 \times 10^6$) isolated from the BM or spleen were incubated with Fc block anti-CD16/32 (Clone 2.4G2) for 10 min at RT. After washing with FACS buffer (0.2% BSA in PBS), cells isolated from the BM were stained with anti-CD19 BUV395 (Clone 1D3), anti-CD138 BV605 (Clone 281-2), and anti-CD45R/B220 PerCP Cy5.5 (Clone RA3-6B2) antibodies for 30 min in 4 °C, protected from light. Cells were washed with FACS buffer and then incubated in Fixation/Permeabilization Solution for 20 min at 4 °C. Upon washing in Wash/Permeabilization solution, cells were stained with anti-IgM FITC (Clone II/41), anti-IgA PE (Clone 11-44-2), and anti-IgG1 AF647 (Thermo Fisher Scientific, A-

21240) for 30 min at 4 °C. After washing, cells were analyzed in terms of calculation intracellular/cytoplasmic intracellular Ig content.

**Intracellular ER staining with for live-cell imaging**. Cells ($1 \times 10^6$) were resuspended in HSBB buffer and then were incubated with 0.75 μl of ER-Tracker™ Red (BODIPY™ TR Glibenclamide, Thermo Fisher Scientific; E34250) for 30 min at 4 °C, in the dark.

**Splenocyte fractionation**. The procedure was performed as described previously[33,54]. Briefly, activated B cells from *Tent5c*-GFP knock-in mice were lysed in MTE buffer (270 mM D-mannitol, 10 mM Tris pH 7.4, 0.1 mM EDTA, 1 mM PMSF) by homogenization monitored by microscopy. Extracts were cleared by sequential centrifugation at 700 and $15,000 \times g$ and subsequently were loaded on the top of the discontinuous sucrose gradient (1.3 M, 1.5 M, and 2 M prepared in 10 mM Tris pH 7.6, 0.1 mM EDTA) and ultracentrifuged at $152,000 \times g$ for 70 min in an SW41 rotor (Beckman). The top layer was collected as cytosol fraction. ER fraction was collected as band at the interphase of 1.3 M sucrose layer, diluted with additional MTE buffer and ultracentrifuged at $126,000 \times g$ for 45 min in MLA130 rotor (Beckman), and pellet was collected as purified ER, resuspended in PBS supplemented with proteases inhibitors and subsequently analyzed with western blot and antibodies against TRAPα (Clone EPR5603), SSR3 (Abcam, ab190936), HDLBP (Bethyl, A303-971A), and PERK (Clone C33E10, GRP94 (Clone H-212), CHOP (Clone D46F1), and GFP (ChromoTek, PABG1-100)).

**ER stress analysis**. WT and *Tent5c* KO B cells were activated with LPS (20 μg/ml) and IL-4 (20 ng/ml) for 3 days. Then, tunicamycin (Sigma) was added to a final concentration of 1 or 2.5 μg/ml for 5 h and cells were collected. RNA was isolated using TRI reagent (Sigma; T9424) according to the manufacturer's instructions and estimations of *Xbp1* splicing and expression level of *Perk*, *Chop*, *Grp94*, *Ero1lB*, and *Ire1α* genes were carried out by RT-qPCR reactions as described above. Cells were also used for protein isolation and western blot analysis of UPR markers levels (XBP1, Clone EPR22004; IRE1, Clone 14C10; PERK, Clone C33E10; ATF6, Clone D4Z8V) as described above.

**Differential expression analysis**. Obtained Illumina sequencing reads were quality filtered using cutadapt (v. 1.18), to remove adapter sequences, low-quality fragments (minimum quality score was set to 20), and too short sequences (threshold set to 30 nt)[55]. Such quality-filtered reads were aligned to the mouse genome (GRCm38, downloaded from the ENSEMBL FTP site, release 94) using the STAR split read aligner (v. 2.6.1a)[56]. Counts for each transcript were collected using featureCounts from Subread package (v. 1.6.3), with options -Q 10 -p -B -C -s 2 -g gene_id -t exon and Gencode vM19 annotation[57]. Statistical analysis of differential expression was performed using the DESeq2 (v.1.22) Bioconductor package[58], using default settings, correcting for the batch effect.

**Poly(A) spike-in generation and evaluation**. mRNA encoding barcoded eGFP with predefined poly(A) tails (A10, A30, A40, A60, A100, and A150) were generated in separate transcription in vitro reaction using SP6 RNA polymerase (NEB) according to the manufacturer's recommendation. mRNA preps were purified using AMPureXP (Beckman). DNA templates for the reaction were a kind gift from Prof. Eivind Valen (University of Bergen) and were described previously[27].

**Nanopore direct RNA sequencing**. Nanopore reads were mapped to GencodeVM22 reference transcript sequences using Minimap 2.16 (ref. [59]). The poly(A) tail lengths for each read were estimated using Nanopolish 0.11.1 polya function[32]. In subsequent analyses, only length estimates with QC tag reported by Nanopolish as PASS were considered. Statistical analysis was performed using functions provided in the NanoTail R package (https://github.com/pbrigf-ibb/nanotail, manuscript in preparation). In detail, we used the Generalized Linear Model approach, with log2(polya length) as a response variable, filtering out transcripts that had a low number of supporting reads in each condition (<20). To correct for the batch effect, a replicate identifier was used as one of the predictors, in addition to the condition (*Tent5c* KO/WT) identifier. Collected $P$ values (for the condition effect) were adjusted for multiple comparisons using the Benjamini–Hochberg method. Transcripts were considered as having poly(A) tail significantly changed between analyzed conditions, if the adjusted $P$ value was < 0.05, the absolute difference in the median tail length was at least 10, and there were at least 20 supporting reads for each condition. All software and algorithms used in this study are listed in Supplementary Table 7.

**Poly(A) spike-ins analysis**. The analysis was done as described previously[27], with minor modifications. Briefly, Nanopore reads were first mapped against the eGFP sequence using minimap 2.16 (ref. [59]), then the eGFP start sequence was located within mapped reads using vMatchPattern from Biostrings. Next, the spike-in barcode was assigned by aligning the expected barcode sequences against the sequence preceding the eGFP start sequence using *pairwise_alignment* function from Biostrings, with type local, match score set to 1, and gapOpening, gapExtension, and mismatch penalties set to 0, −1, and −1, respectively. The alignment

score was normalized by dividing by the query length. The barcode with the highest normalized alignment score (and above a threshold of 0.3) was assigned to the read. Poly(A) lengths were estimated for all eGFP mapped reads with Nanopolish 0.11.1 polya function and assigned to given spike-in based on barcode.

**Statistics and reproducibility**. Group size was based on the previous experience. No statistical method was used to predetermine sample size. Statistical analysis was conducted on data from three or more biologically independent experimental replicates. Statistical analysis of quantitative data was performed using Prism 6 software (GraphPad) unless otherwise stated. The statistical tests used in each instance are mentioned in the figure legends. All data were checked for normality using the Shapiro–Wilk test. Data are presented as scatter dot plots or bar plots, with mean values indicated and standard deviation shown as the error bars, as indicated in the figure legends, and individual data points are shown. Unless indicated otherwise "$n$" represents the number of animals of each genotype used in a study. Most of the experiments were repeated at least twice, leading to comparable results with exception of RNase H assay, which was repeated once. The exact number of biological replicates used for statistical analysis is stated for every single experiment. Samples with clear technical failures during tissue harvesting, cells isolation, processing, or data collection throught flow cytometry or biochemical experiments were excluded from analyses.

## Data availability
mRNA expression data are deposited in GEO with the accession code GSE132883. Nanopore direct RNA sequencing are deposited at European Nucleotide Archive with the accession code PRJEB33089. Raw data underlying figures are provided as a Source Data file or in Supplementary Fig. 8 and Supplementary Datasets, or are also available from the corresponding authors upon reasonable request.

## Code availability
All custom code used in this study is deposited at https://github.com/pbrigf-ibb/nanotail.

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

## Acknowledgements

We would like to thank Maximilian Krause and Eivind Valen (University of Bergen) for DNA templates encoding ONT spike-ins, Aleksander Chlebowski for microscopy assistance, Dominik Cysewski for high-resolution MS analyses, Marcin Szpila for assistance in coordination of animal housework, Agnieszka Tudek for providing RNA from HEK293T cell line and 4-thiouracil-labeled *S. pombe* cells, and Janina Durys for the editing of the manuscript and all laboratory members for stimulating discussions. The equipment from www.ibb.waw.pl/en/services/mass-spectrometry-lab was sponsored in part by the Centre for Preclinical Research and Technology (CePT), a project co-sponsored by European Regional Development Fund and Innovative Economy, The National Cohesion Strategy of Poland. This work was funded by NCN OPUS14, UMO-2017/27/B/NZ2/01234 (to S.M.) and co-supported by NCN Harmonia10 UMO-2013/10/M/NZ4/00299 (to A.D.), FNP TEAM TECH CORE FACILITY/2017-4/5 (to A.D.), and "Regional Initiative of Excellence" program by the Ministry of Science and Higher Education 013/RID (to J. Golab).

## Author contributions

S.M. and A.D. developed and directed the studies, A.B and S.M. carried out the majority of the biochemical and cell line experiments, M.K.-K. performed all flow cytometry analyses, P.S.K. performed all bioinformatics analyses, O.G. analyzed mice blood, performed non-immunized serum ELISA tests, participated in localization studies, and coordinated work of the animal house, B.T. performed tissue immunostaining, K.K. purified the eIF4E protein, D.N. and J. Golab immunized mice and performed immunized serum ELISA tests, J. Gruchota prepared CRISPR reagents and genotyped mice, and E.B. generated transgenic animals. S.M., A.D., P.S.K., A.B., O.G., and M.K.-K.wrote the manuscript.

## Competing interests

The authors declare no competing interests.
