## [Peer Review File · Nature Communications]

Reviewers' comments:

Reviewer #1 (Remarks to the Author):

In this study, Bilaska et al presented the B cell phenotype upon depletion of TENT5C, a non-canonical cytoplasmic poly(A) polymerase. Although the observed phenotype of impaired immunoglobulin production is interesting, these are descriptive observations overall based mainly on in vitro study, and the authors failed to provide cause-and-effect relationship for the molecular mechanism explaining the observed phenotype.

Major point;

IgG reduction observed in TENT5C KO mice could be a result of 1. impaired B cell proliferation or plasma cell differentiation, 2. impaired class-switching (Fig. 2, Fig. 6), 3. impaired ER function (Fig. 7), or 4. impaired immunoglobulin transcription and/or translation caused by shortened poly(A) tail (Fig.1). The authors did not provide the evidence for cause-and-effect relationship between TENT5C-deficiency and observed phenotype.

Minor points;

1. IgG κ or IgG λ mean the Ig κ or Ig λ light chain paired with IgG heavy chain? The authors should clarify this ambiguous description.
2. Fig. 2d, Fig. 6a etc.; What is the IgG1+IgM+ double positive population? It is quite unusual that such population is observed as major population. The method should be re-examined carefully.
3. Fig. 3b shows that TENT5C expression is not restricted to CD138+ plasma cells, but starts at activated B cells before expressing CD138, which should be correctly described.
4. Increased spleen mass (Fig. 5a) may not simply suggest the enhanced proliferation of TENT5C-deficient B cells. How about the T cell population in KO mice?

Reviewer #2 (Remarks to the Author):

The paper contains very interesting data on poly(A) tail regulation of immunoglobulin mRNAs by TENT5, using a new high throughput method and giving important data on the role of TENT5 in B cell development and immunoglobulin production. As a relatively novel poly(A) polymerase family, TENTs have a high novelty value.

However, the paper lacks proof of the effect of a lack of TENT5C on immunoglobulin mRNA stability and the validation by northern blot needs oligodT/RNase H deadenylated control samples to show the size difference is truly due to a poly(A) tail.

The evidence for a direct role of TENT5C would be much stronger if it could be shown that the mRNAs that change poly(A) tail size are also bound by TENT5, but this could be a problem for a transient interaction, as is common with poly(A) polymerases.

The claim that TENT5 is localised on the ER is not justified by the data, in my opinion, and further evidence needs to be included if this claim is to be published (details below).

Detailed comments:

Nanopore poly(A) sequencing relies on the estimation of the transit time per base. Small errors in this estimation will disproportionately affect longer poly(A) tails. As apparently no spike-ins were used, it is hard to check how well the measured sizes correlate to real sizes in this experiment. The northern blot shown to validate the Nanopore data lacks the key control of oligodT/RNaseH treated RNA to remove the poly(A) tail and markers to allow an estimate of the size difference detected by this method. A simple PCR based poly(A) test may also allow confirmation of the poly(A) size change (sequencing of the product can confirm that there is no alternative polyadenylation).

P4 paragraph ending line 126. State in the text how much starting material was required for this method. This is important for the reader to assess the practicality of the method for their purposes.

Line 127: I presume 1.5M means 1.5 million (10⁶). This is not a standard abbreviation, amend. Are these reads for both or for one set of samples? It would better to indicate in the text that duplicates were performed. Information is also missing on how many reads did not give poly(A) tails and how many nanopore modules were required to obtain the number of reads.

Paragraph starting line 127. "which provided reliable information about steady-state poly(A) tail length" A key problem in all methods of poly(A) tail sequencing is that a large depth is required to get information on rarer mRNAs. In this paragraph, indicate for how many species of mRNA a poly(A) tail size distribution and/or median size could be assigned with confidence (2790 listed in the supplementary data) and what criteria were used to select these. How were transcript variants handled?

It would be good to have information about the variability of for each mRNA species in the supplementary data, rather than just the mean, geometric mean and median values. As these numbers will be available for both replicates, it's not clear to me how the two replicates are represented in these data. Plots of the median values of the replicates against each other would also show how close the two replicates are.

The authors claim that TENT5C is ER associated, but I do not find the data convincing. In the western in Figure 7a, Flag-TENT5C, the non-specific Flag bands and the cytoplasmic marker alpha tubulin are all similarly distributed. This looks like cytoplasmic localisation to me. The microscopy in Figure 7b is of insufficient magnification to allow a conclusion of co-localisation at a subcellular scale. Ideally, colocalization should be quantified from images at a higher magnification.

The authors make much of the effect of TENT5C knockout on ER stress, but then do admit that this is likely to be a consequence of the reduced immunoglobulin mRNA levels. In my opinion, this is not a finding that merits the "importantly" in the abstract.

The paper skirts around the key issue of how TENT5C knockout causes a reduction in immunoglobulin mRNA levels. Is TENT5C mediated polyadenylation required to stabilize the mRNAs? A relatively simple experiment inhibiting transcription with actinomycin D could confirm this.

To be fully convincing mechanistically, evidence of an interaction of TENT5C with the immunoglobulin mRNAs would be desirable.

As the paper is on the mouse gene, should the gene name not be *Tent5c* (in italics)?

Reviewer #3 (Remarks to the Author):

In this study, the authors present observations concerning molecular and organismal phenotypes of TENT5C knockout mice. The molecular findings center on shortening of the poly(A)-tails of the immunoglobulin (Ig) mRNAs, observed by Nanopore sequencing technology. This shortening coincides with decreased Ig expression level. At the cell biological and organismal level, the authors report decreased spleen size, lower antibody secretion, and phenotypes associated with ER volume. Overall, this study seems to present an interesting set of observations concerning the role of TENT5C, although there are some technical concerns that must be addressed.

Major points:

1. The authors make several claims about how the ONT technology compares to Illumina's and is less prone to biases. While this is indeed true for any bias due to RT and PCR, ONT has other issues that may make it more error-prone (Krause et al. 2019, bioRxiv tailfindr: Alignment-free poly(A) length measurement for Oxford Nanopore RNA and DNA sequencing, and Workman et al. 2018 bioRxiv Nanopore native RNA sequencing of a human poly(A) transcriptome). Furthermore, the authors make no effort to show that their technique robustly measures tails. Without standards, the authors can only say that their technique measures relative tail lengths, not absolute ones, and while this may not impact the study, it does impact the claims made concerning ONT technology and tail-length measurements.
2. The claim that TENT5C is ER bound is not supported (Figure 7). Regarding panel a, the authors suggest that a significant fraction of TENT5C is membrane bound. However, the relative abundance of TENT5C in the cytoplasm and membrane fractions closely mirrors that of alpha-tubulin, rather than that of GRP94, which indicates that TENT5C is predominantly cytoplasmic, and signal for co-fractionation with membrane could easily be cytoplasmic contamination. The authors should also show quantitation of the western blot. Regarding Fig. 7b, from the single TENT5C-GFP cell provided, GFP does not appear to co-localize with ER Tracker. In order to better evaluate whether TENT5C-GFP co-localizes with ER, the authors should also include a GFP fusion with a cytosolic protein (or GFP alone). In addition, the authors should include images from more than a single cell for each condition, and provide a quantitation of the co-localization. Regarding Fig. 7c, the authors should also quantify the ER content of the TENT5C-GFP knock-in, as the ER seems to be expanded in the TENT5C-GFP knock-in shown in 7b.
3. A major claim of the paper is that tail shortening in the mutant is specific to Ig mRNAs. However, this claim is not clearly supported. The authors show that the global distribution of tails (and the distribution of tails for mRNAs for ribosome proteins) does not change. However, analysis of the mean tail lengths in their spreadsheet (Table P3J6DX) indicates that there is little correspondence between tail lengths with and without TENT5C ($r^2 = 0.25$). Therefore, tails of many other mRNAs appear to be changing. This r^2 improves with cutoffs that exclude mRNAs with fewer read counts. However, this also excludes many genes whose tails might be changing. To provide evidence that the tail shortening is specific, the authors should show a scatter plot of mean tail lengths in WT vs KO, choosing a cutoff that supports their claim but also acknowledging that a major disadvantage of their method compared to the Illumina methods is that it cannot accurately speak to tail lengths of as many genes.

Minor points:

1. Fig 1. The northern blot is not quantified and has no ladder—what is the change in tail length as determined by the northern, and how does that compare to change observed in the ONT data?
2. The authors mention that the reverse transcription step of the tail-length sequencing protocol is optional and do not clearly state whether this step was performed to generate the data shown in the paper. The authors should indicate whether or not this step was performed.
3. In lines 141–143, the authors refer to “correlation scatterplot between the poly(A) tail length change and expression fold change (Supplementary Fig. 1h)” however Supp. Fig. 1h does not show fold changes, but instead poly(A) tail length and expression of only WT.
4. The authors should indicate what levels of significance the asterisks denote.
5. In figure 1G, and elsewhere, it is unclear what the bars represent (SEM, SD, or confidence

interval).

6. Also in Fig. 1G, what is the relationship between the qPCR fold-change and the fold-change from Fig. 1E? Additionally, the authors should add the number of immunoglobulin genes in Fig. 1E and 1F in the key.

7. In lines 153–159, the authors refer to immunoglobulin transcripts as having longer poly(A) tails compared to *other* highly abundant mRNAs, but on average, immunoglobulin mRNAs are not highly abundant compared to the other mRNAs examined (Fig. Supp. 1h). Because they are not highly abundant as a class, their tails lengths tail lengths can't be used to refute the claim that highly abundant mRNAs tend to have shorter tails. Indeed, because this previously claim is merely for a tendency and not a strict rule, a few exceptions to this tendency cannot "oppose previous reports" of the tendency.

8. In lines 179–184, the authors refer to the effect being specific since total serum protein concentrations, as well as a several sub fractions, are unchanged. However, contrarily, alpha globulins are elevated in the TENT5C KO, which is inconsistent with the conclusion that TENT5C KO leads to decreased immunoglobulin synthesis in mice.

9. In lines 195–197, the authors refer to TENT5C as "mainly expressed in the population of CD138-positive cells." While it is true that nearly all CD138 positive cells are also TENT5C positive, fewer than 50% of TENT5C positive cells are CD138 positive (Fig. 3B) (though TENT5C expression is higher in CD138 cells compared to CD138 negative cells).

Point by point response to the reviewer's comments

We thank the Reviewers for a positive opinion about our paper. We address all issues raised by reviewers and edited the manuscript to improve its clarity. Below we provide a point by point response to the Reviewer's comments.

Reviewers' comments:

Reviewer #1 (Remarks to the Author):

In this study, Bilaska et al presented the B cell phenotype upon depletion of TENT5C, a non-canonical cytoplasmic poly(A) polymerase. Although the observed phenotype of impaired immunoglobulin production is interesting, these are descriptive observation overall based mainly on *in vitro* study, and the authors failed to provide cause-and-effect relationship for the molecular mechanism explaining the observed phenotype.

We wish to thank the Reviewer for this insightful comment. We agree that the original version of the manuscript insufficiently addressed full molecular mechanisms and the impact of immunoglobulin mRNAs polyadenylation by TENT5C *in vivo*.

In the revised version of the manuscript, we show that indeed, *Tent5C* KO leads to decreased humoral response to model antigens *in vivo*. Moreover, in order to strengthen the mechanistic aspects, we analyzed newly generated catalytically inactive *Tent5C knock-in* mice and show that the phenotypes recapitulate the data for KO, which is a very strong indication that the activity of this enzyme is essential for all its functions.

Major point;

IgG reduction observed in TENT5C KO mice could be a result of 1. impaired B cell proliferation or plasma cell differentiation, 2. impaired class-switching (Fig. 2, Fig. 6), 3. impaired ER function (Fig. 7), or 4. impaired immunoglobulin transcription and/or translation caused by shortened poly(A) tail (Fig.1). The authors did not provide the evidence for cause-and-effect relationship between TENT5C-deficiency and observed phenotype.

As mentioned above, the phenotype of the TENT5C catalytic mutant indicates that the function is related to its activity. Moreover, Ig mRNAs have shorter poly(A) tails and are less stable (new data). Taking all these into consideration, we find it difficult to imagine that the observed phenotypes are not, to a large extent, related to TENT5C mediated stabilization of Ig mRNAs.

Minor points;

1. IgG κ or IgG λ mean the Ig κ or Ig λ light chain paired with IgG heavy chain? The authors should clarify this ambiguous description.

The description was changed.

2. Fig. 2d, Fig. 6a etc.; What is the IgG1+IgM+ double positive population? It is quite unusual that such population is observed as major population. The method should be re-examined carefully.

The data were carefully reanalyzed, we especially optimized the compensation to minimize potentially misleading double-positive population. In the new version of the manuscript, we focused on the IgG1 population only.

It should be, however, noticed that our *in vitro* experiments in which B cell mature are in a relatively short time scale. Thus, the putative double-positive population may simply reflect cells that have already undergone class switch recombination, but the mRNA and protein products of IgM are still temporary present. Nevertheless, we decided to not discuss this issue in the manuscript, which focuses on polyadenylation rather than class-switch recombination.

3. Fig. 3b shows that TENT5C expression is not restricted to CD138+ plasma cells, but starts at activated B cells before expressing CD138, which should be correctly described.

We noticed that the expression of TENT5C-GFP is highest in the CD138+ cells at day 10 of activation, suggesting that the peak of TENT5C expression falls into the plasma cells. Although expression of TENT5C begins before cells develop into the plasma cells, only plasma cells are nearly 100% GFP-positive, which suggests an important role of this enzyme at the last steps of B cell differentiation.

4. Increased spleen mass (Fig. 5a) may not simply suggest the enhanced proliferation of TENT5C-deficient B cells. How about the T cell population in KO mice?

We agree that there could be many reasons for spleen mass expansion. One of them can be a faster proliferation of T cells however, analysis of T cells subpopulations in the spleen was out of the scope of this paper. Importantly, TENT5C is a negative regulator of B cell proliferation and differentiation.

Reviewer #2 (Remarks to the Author):

The paper contains very interesting data on poly(A) tail regulation of immunoglobulin mRNAs by TENT5, using a new high throughput method and giving important data on the role of TENT5 in B cell development and immunoglobulin production. As a relatively novel poly(A) polymerase family, TENTs have a high novelty value.

Thank you for a positive opinion about our paper.

However, the paper lacks proof of the effect of a lack of TENT5C on immunoglobulin mRNA stability and the validation by northern blot needs oligo dT/RNase H deadenylated control samples to show the size difference is truly due to a poly(A) tail.

All requested controls including RNaseH/oligo(dT)₂₀ are now included (Fig.1h and 8a) in the revised version of the manuscript. They confirm that mRNA heterogeneity indeed results from 3'-end polyadenylation.

The evidence for a direct role of TENT5C would be much stronger if it could be shown that the mRNAs that change poly(A) tail size are also bound by TENT5, but this could be a problem for a transient interaction, as is common with poly(A) polymerases.

We agree that showing the direct interaction of TENT5C would strongly support our manuscript.

To address this issue, we used the RNA immunoprecipitation (IP) approach after UV or formaldehyde crosslinking of activated B cells from *Tent5C*-GFP. Unfortunately, using both strategies we were not able to show enrichment of immunoglobulin mRNAs after TENT5C IP from B cells. However, the CLIP-based approach was successfully used by our laboratory to show TENT5C (FAM46C) interaction with RNA in HEK293 cells but in this case it was strongly overexpressed (Mroczek et al 2017; *Nat Commun*).

Thus, it seems that the TENT5C interactions with substrates are very transient, which is consistent with the lack of any detectable RNA-binding domains in these proteins. Further research is needed to uncover mechanistic details of substrate recognition by TENT5C protein.

However, our new data, showing that Ig mRNAs isolated from catalytically inactive TENT5C mutant mice also have shorter poly(A) tail support the direct involvement of TENT5C in Ig mRNAs polyadenylation.

The claim that TENT5 is localised on the ER is not justified by the data, in my opinion, and further evidence needs to be included if this claim is to be published (details below).

ER paragraph has been rewritten and new experiments were added (Fig. 7a).

Detailed comments:

Nanopore poly(A) sequencing relies on the estimation of the transit time per base. Small errors in this estimation will disproportionately affect longer poly(A) tails. As apparently no spike-ins were used, it is hard to check how well the measured sizes correlate to real sizes in this experiment. The northern blot shown to validate the Nanopore data lacks the key control of oligodT/RNaseH treated RNA to remove the poly(A) tail and markers to allow an estimate of the size difference detected by this method. A simple PCR based poly(A) test may also allow confirmation of the poly(A) size change (sequencing of the product can confirm that there is no alternative polyadenylation).

In the revised manuscript we have included data from the Nanopore sequencing of the additional replicate, with spike-ins included (the same as in Krause et al., 2019). The analysis confirmed the accuracy of the method, and the distribution of poly(A) length was consistent with the previous report (especially bimodal distribution for longer spike-ins, visible on the Supplementary Figure 1g in the revised manuscript, for the explanation of possible reasons please see Krause et al., 2019). Additionally, we provide data indicating that the accuracy of the predictions can also be assessed with mitochondrial transcripts (Supplementary Figure 1i), which have well-defined poly(A) tails. In fact comparison of both, synthetic spike-ins and

mitochondrial transcripts, shows the advantage of direct RNA sequencing, as distribution of predicted poly(A) lengths for mt-Co2 transcript forms a much sharper peak than for any of spike-ins (probably due to processing steps during spike-ins preparation, like PCR amplification and *in vitro* transcription).

Finally, the Nanopore data were also validated *in vitro*, as requested, using oligo(dT)₂₀/RNaseH treatment.

P4 paragraph ending line 126. State in the text how much starting material was required for this method. This is important for the reader to assess the practicality of the method for their purposes.

Such information is now included in the revised version of the manuscript. Materials and methods section concerning sequencing was rewritten and now is more detailed.

Line 127: I presume 1.5M means 1.5 million (106). This is not a standard abbreviation, amend. Are these reads for both or for one set of samples? It would be better to indicate in the text that duplicates were performed. Information is also missing on how many reads did not give poly(A) tails and how many nanopore modules were required to obtain the number of reads.

This section was rewritten, and all requested pieces of additional information are included. The number was a sum of all mapped reads for all samples analyzed. Each sample was sequenced on a single MinION flowcell (R9.4.1 pore). In the revised version we have added another replicate and we provide the table (Supplementary Table 1) summarizing all sequencing runs.

Paragraph starting line 127. "which provided reliable information about steady-state poly(A) tail length" A key problem in all methods of poly(A) tail sequencing is that a large depth is required to get information on rarer mRNAs. In this paragraph, indicate for how many species of mRNA a poly(A) tail size distribution and/or median size could be assigned with confidence (2790 listed in the supplementary data) and what criteria were used to select these. How were transcript variants handled?

As the sequencing depth possible with Nanopore is limited, especially when a low amount of RNA material is available (as in the case of B cells isolated from a single animal), we are aware that we can miss some less abundant RNA species. However, the method is easily scalable, and if enough starting material is available, up to 4 million direct RNA reads per MinION flowcell can be obtained (we routinely achieve 2-3.5 million reads per flowcell in other sequencing experiments). Thus, constant improvement in the technology should allow poly(A) sequencing of less abundant transcripts.

In the revised version of the manuscript, 3 biological replicates of each condition were analyzed (2 in the original version, the number of reads increased from 1.5 to 2.5 million). The poly(A) length shifts for lgs were observed in each pair.

Initially, we arbitrarily decided to calculate statistics only for genes that got at least 10 reads in any of the conditions. In our opinion, this was a fair compromise, allowing for the analysis of a significant number of abundant transcripts, and limiting false hits originating from internal variability of poly(A) measurement (Figure R1, below). In the revised manuscript, thanks to the increased depth of sequencing and the third replicate, we set the threshold to at least 20 reads in both WT and KO datasets. This allowed computing statistics for 4264 genes (Supplementary

Dataset 1), which is sufficient to draw some general conclusions regarding the poly(A) tail distribution genome-wide.

Transcript variants were not examined in this manuscript. It is difficult to precisely map, especially 5' isoforms since it is frequent with Nanopore sequencing that 5' ends are not entirely sequenced. Therefore all reads mapped to the transcripts originated from the single locus (gene) were analyzed together. However, in the reference used for mapping, all possible isoforms were included to increase mapping efficiency.

Figure R1. The correlation coefficient (Pearson's r) of poly(A) mean lengths per gene between TENT5C_WT and TENT5C_KO for different count thresholds.

It would be good to have information about the variability of for each mRNA species in the supplementary data, rather than just the mean, geometric mean and median values. As these numbers will be available for both replicates, it's not clear to me how the two replicates are represented in these data. Plots of the median values of the replicates against each other would also show how close the two replicates are.

There was some variability between replicates (Figure R2). One of the reasons is the batch effect (different mice batches were used for experiments) Another reason is the fact that poly(A) lengths were estimated for each read individually, despite the natural heterogeneity of transcriptome. However, for each pair (WT + KO) compared, there was a consistent distribution of observed poly(A) lengths. We have added additional plots in the Supplement to show the data consistency (Supplementary Figure 1f). Also, the Supplementary dataset contains more detailed data, with mean and variability values for all replicates.

Figure R2. Correlation scatterplot comparing mean poly(A) lengths in all sequencing runs for each transcript. Only transcripts with at least 20 reads for each condition are shown. In the upper right panel Pearson's r values are shown.

The authors claim that TENT5C is ER associated, but I do not find the data convincing. In the western in Figure 7a, Flag-TENT5C, the non-specific Flag bands and the cytoplasmic marker alpha tubulin are all similarly distributed. This looks like cytoplasmic localisation to me. The microscopy in Figure 7b is of insufficient magnification to allow a conclusion of co-localisation at a subcellular scale. Ideally, colocalization should be quantified from images at a higher magnification.

Thank you for that comment. We agree that „ER-associated“ was an exaggerated term. We have used different state-of-the-art experimental approaches for ER isolation and slightly changed the final conclusions. A minor fraction of TENT5C is still detectable in pure ER isolated from activated B cells. However, the majority of the protein is cytoplasmic which is characteristic of ncPAPs. We suppose that TENT5C can be associated “temporarily” with ER if it is needed especially if it results from physiological conditions during which ER or ER-related functions are of particular importance. Moreover, TENT5C is not part of the stable protein complex. Thus, the mechanism of co-purification with the ER remains to be established.

Finally, analysis of the ER volume in B cells isolated from TENT5C Cat mice showed decreased expansion dynamics during activation comparing to WT littermates similar to the one we have demonstrated for KO mice. This proves that TENT5C catalytic activity shaping B cells transcriptome affects ER function.

The manuscript was modified accordingly.

The authors make much of the effect of TENT5C knockout on ER stress, but then do admit that this is likely to be a consequence of the reduced immunoglobulin mRNA levels. In my opinion, this is not a finding that merits the “importantly” in the abstract.

This has been changed.

The paper skirts around the key issue of how TENT5C knockout causes a reduction in immunoglobulin mRNA levels. Is TENT5C mediated polyadenylation required to stabilize the mRNAs? A relatively simple experiment inhibiting transcription with actinomycin D could confirm this.

Analysis of half-life for selected mRNAs targeted by TENT5C revealed that polyadenylation stabilizes them. Two new panels (i and j) were added to Fig 1.

To be fully convincing mechanistically, evidence of an interaction of TENT5C with the immunoglobulin mRNAs would be desirable.

We agree with Reviewer 2 and Reviewer 3 that showing directly the interaction of TENT5C would strongly support our manuscript.

To address this issue, we used the RNA immunoprecipitation approach after UV or formaldehyde crosslinking of activated B cells from TENT5C-GFP. Unfortunately, using both strategies we were not able to show enrichment of immunoglobulins mRNA after TENT5C IP. Moreover, the CLIP-based strategy was successfully used by our laboratory to show TENT5C (FAM46C) interaction with RNA in HEK293 cells (Mroczek et al 2017; *Nat Commun*).

Thus, it seems that the TENT5C interactions with substrates are very transient, which is consistent with the lack of any detectable RNA-binding domains in these proteins. Further research is needed to uncover mechanistic details of substrate recognition by TENT5C protein.

As the paper is on the mouse gene, should the gene name not be *Tent5c* (in italics)?

Nomenclature has been changed.

Reviewer #3 (Remarks to the Author):

In this study, the authors present observations concerning molecular and organismal phenotypes of TENT5C knockout mice. The molecular findings center on shortening of the poly(A)-tails of the immunoglobulin (Ig) mRNAs, observed by Nanopore sequencing technology. This shortening coincides with decreased Ig expression level. At the cell biological and organismal level, the authors report decreased spleen size, lower antibody secretion, and phenotypes associated with ER volume. Overall, this study seems to present an interesting set of observations concerning the role of TENT5C, although there are some technical concerns that must be addressed.

We thank the Reviewer for a positive opinion about our paper.

Major points:

1. The authors make several claims about how the ONT technology compares to Illumina's and is less prone to biases. While this is indeed true for any bias due to RT and PCR, ONT has other issues that may make it more error-prone (Krause et al. 2019, bioRxiv tailfindr: Alignment-free poly(A) length measurement for Oxford Nanopore RNA and DNA sequencing, and Workman et al. 2018 bioRxiv Nanopore native RNA sequencing of a human poly(A) transcriptome). Furthermore, the authors make no effort to show that their technique robustly measures tails. Without standards, the authors can only say that their technique measures relative tail lengths, not absolute ones, and while this may not impact the study, it does impact the claims made concerning ONT technology and tail-length measurements.

The part of the manuscript which compares ONT and Illumina technologies was changed. Additionally, we made another sequencing runs, with the introduction of standards with pre-defined poly(A) lengths, showing that obtained predictions are valid. Our results are consistent with observations from articles mentioned by the Reviewer, showing that Nanopore-based measurements are reproducible. We have also included *in vitro* validation using Northern Blot with oligo(dT)₂₀/RNase H treated samples. As it was mentioned in response to Reviewer 2, in the new version of the manuscript we also show the poly(A) lengths distribution for mitochondrial transcripts (mt-Co2, shown in Supplementary Fig. 1i, but this is valid also for the other mitochondrial transcripts) All new data and analysis indicate that predicted lengths are consistent with the knowledge in the field, and that poly(A) lengths distribution forms much sharper peak that in the case of synthetic spike-ins.

2. The claim that TENT5C is ER bound is not supported (Figure 7). Regarding panel a, the authors suggest that a significant fraction of TENT5C is membrane bound. However, the relative abundance of TENT5C in the cytoplasm and membrane fractions closely mirrors that of alpha-tubulin, rather than that of GRP94, which indicates that TENT5C is predominantly cytoplasmic, and signal for co-fractionation with membrane could easily be cytoplasmic contamination. The authors should also show quantitation of the western blot.

Thank you for this comment. We agree that simple kit-based purification does not provide enough confidence for our statement, thus we have changed the ER purification strategy. This figure, as well as final conclusions, were changed.

Regarding Fig. 7b, from the single TENT5C-GFP cell provided, GFP does not appear to co-localize with ER Tracker. In order to better evaluate whether TENT5C-GFP co-localizes with ER, the authors should also include a GFP fusion with a cytosolic protein (or GFP alone). In addition, the authors should include images from more than a single cell for each condition, and provide a quantitation of the co-localization.

Thank you for this comment. We have changed the microscopy panel. Moreover, the results of new fractionation-based biochemical experiments showed that only a minor fraction of TENT5C protein can be detected on ER. We decided to use ER tracker since all transfection/transduction-based techniques provide rather high overexpression levels, which not complies with colocalization studies. We currently do not have mice expressing a protein fused to a fluorescent tag and located in the ER or cytoplasm only that would be expressed in B cells at a physiological level. Thus we decided to use ER tracker dye.

Regarding Fig. 7c, the authors should also quantify the ER content of the TENT5C-GFP knock-in, as the ER seems to be expanded in the TENT5C-GFP knock-in shown in 7b.

This has been done. We analyzed ER size in B cells isolated from *Tent5C*-GFP knock-in mice using dye tracker staining followed by FACS analysis and there is no difference. See the graph below.

Moreover, other phenotypes observed in *Tent5C* KO or *Tent5C* Cat mice such as an increased number of CD138+ cells or accelerated activation were not observed in *Tent5C*-GFP knock-in mice. Please see the two graphs below.

3. A major claim of the paper is that tail shortening in the mutant is specific to Ig mRNAs. However, this claim is not clearly supported. The authors show that the global distribution of tails (and the distribution of tails for mRNAs for ribosome proteins) does not change. However, analysis of the mean tail lengths in their spreadsheet (Table P3J6DX) indicates that there is little correspondence between tail lengths with and without TENT5C ($r^2 = 0.25$). Therefore, tails of many other mRNAs appear to be changing. This r^2 improves with cutoffs that exclude mRNAs with fewer read counts. However, this also excludes many genes whose tails might be changing. To provide evidence that the tail shortening is specific, the authors should show a scatter plot of mean tail lengths in WT vs KO, choosing a cutoff that supports their claim but also acknowledging that a major disadvantage of their method compared to the Illumina methods is that it cannot accurately speak to tail lengths of as many genes.

As it was mentioned in response to Reviewer 2, due to the limited sequencing depth of the Nanopore system, it is likely that some less abundant RNA species with only a few reads for them present in the dataset, are under-represented in the analysis. Natural heterogeneity (the fact that poly(A) length for given transcript is not set with a single-nucleotide precision and transcripts at different adenylation/deadenylation stages can be recovered during library preparation step) may cause biased sampling of poly(A) lengths of such low abundant transcripts and lead to false conclusions about the mean poly(A) length and lower correlation between samples. Thus, we arbitrarily decided to calculate statistics only for the genes that got at least 20 (10 in the original manuscript) reads in any of the conditions. In our opinion, this was a fair compromise, allowing for the analysis of enough number of abundant transcripts, and limiting false hits, probable with a low number of reads. This was supported by the analysis of the correlation coefficient between WT and KO samples with increasing counts threshold for single RNA species (Figure R1, in the response to Reviewer 2). With the threshold set to at least 20 reads per transcript Pearson's r is 0.74, and 4264 genes can be analyzed with confidence.

Minor points:

1. Fig 1. The northern blot is not quantified and has no ladder—what is the change in tail length as determined by the northern, and how does that compare to change observed in the ONT data?

All requested controls including size markers and loading control are now included (Fig. 1h, Fig. 8a and Supplementary Figure 8). ONT data were validated by the addition of mRNAs with predefined poly(A) tail length (Supplementary Fig. 1g).

2. The authors mention that the reverse transcription step of the tail-length sequencing protocol is optional and do not clearly state whether this step was performed to generate the data shown in the paper. The authors should indicate whether or not this step was performed.

Such information is now included in the revised version of the manuscript. Materials and methods section concerning sequencing was rewritten and now is more detailed.

3. In lines 141–143, the authors refer to “correlation scatterplot between the poly(A) tail length change and expression fold change (Supplementary Fig. 1h)” however Supp. Fig. 1h does not show fold changes, but instead poly(A) tail length and expression of only WT.

Thank you for noticing that. It was corrected.

4. The authors should indicate what levels of significance the asterisks denote.

Such information is now provided in the Methods section, in the Quantification and statistical analysis paragraph.

5. In figure 1G, and elsewhere, it is unclear what the bars represent (SEM, SD, or confidence interval).

Such information is now provided in the Methods section, in the Quantification and statistical analysis paragraph. Unless otherwise stated, error bars represent standard deviation.

6. Also in Fig. 1G, what is the relationship between the qPCR fold-change and the fold-change from Fig. 1E? Additionally, the authors should add the number of immunoglobulin genes in Fig. 1E and 1F in the key.

The numbers are provided in the revised manuscript. qPCR analysis, shown in Figure 1g, was intended to validate RNAseq differential analysis hits (Fig. 1f), with decreased poly(A) tails observed with the Nanopore sequencing. Fold change values obtained with both, qPCR and DESeq2 analysis, were consistent, and followed the same trend (see table below, showing fold changes between *Tent5C* KO and WT for 3 methods used for transcripts quantification). For the Nanopore the changes were less pronounced (what can be seen also on Fig. 1e), but for two of four tested transcripts, they were consistent with Illumina and qPCR results. This may originate from the fact of low sequencing depth in the case of Nanopore.

	Illumina	qPCR	Nanopore
Igkc	0,536	0,630	0,731
Igk1	0,792	0,711	0,856
Jchain	0,728	0,506	1,018
Ighm	0,783	0,734	1,018

7. In lines 153–159, the authors refer to immunoglobulin transcripts as having longer poly(A) tails compared to *other* highly abundant mRNAs, but on average, immunoglobulin mRNAs are not highly abundant compared to the other mRNAs examined (Fig. Supp. 1h). Because they are not highly abundant as a class, their tails lengths tail lengths can't be used to refute the claim that highly abundant mRNAs tend to have shorter tails. Indeed, because this previously claim is merely for a tendency and not a strict rule, a few exceptions to this tendency cannot "oppose previous reports" of the tendency.

This paragraph has been rewritten. For both Illumina and Nanopore sequencing, the cumulative abundance of all immunoglobulin-assigned reads is 2.5% and 2.1%, respectively (in relation to all annotated reads), what, in our opinion, makes them highly abundant as a class. This is consistent with the major function of B cells. Mean poly(A) length of immunoglobulin transcripts are also indeed longer than for other highly expressed genes (Figure R3). However, we agree that this phenomenon should be treated as the exception of the tendency shown in other reports.

Figure R3. Comparison of mean tail lengths for: immunoglobulins, ribosomal proteins, and (out of remaining) 100 most expressed transcripts.

8. In lines 179–184, the authors refer to the effect being specific since total serum protein concentrations, as well as several subfractions, are unchanged. However, contrarily, alpha globulins are elevated in the *TENT5C* KO, which is inconsistent with the conclusion that *TENT5C* KO leads to decreased immunoglobulin synthesis in mice.

We have reworded the sentence to make it less ambiguous. Additionally, we confirm decreased gamma globulins concentrations also in *Tent5C* Cat mice (Fig. 8f) and by ELISA test for 5 different immunoglobulins for *Tent5C* KO (Fig. 2h). Our statement is also consistent with the results of immunizations with model antigens (Fig. 2i, j).

9. In lines 195–197, the authors refer to *TENT5C* as “mainly expressed in the population of CD138-positive cells.” While it is true that nearly all CD138 positive cells are also *TENT5C* positive, fewer than 50% of *TENT5C* positive cells are CD138 positive (Fig. 3B) (though *TENT5C* expression is higher in CD138 cells compared to CD138 negative cells).

This has been corrected.

Reviewers' comments:

Reviewer #1 (Remarks to the Author):

In the revised version, biochemical part is very good, but author still did not address my concerns. Indeed, in the response letter, authors mentioned that they find it difficult to the observed phenotypes are not, to a large extent, related to TENT5C mediated stabilization of Ig mRNA.

Reviewer #2 (Remarks to the Author):

The manuscript is much improved and I'd like to thank the authors for their excellent work. The data from the catalytic mutant mouse are an especially good addition that adds to the impact of this paper. The work is nearly ready for publication in my opinion.

I have just a few minor comments that need further clarification:

In my opinion, the authors misuse the word "explicitly" in the abstract. While explicit does mean "clear, exact, unambiguous", it is only used to refer to human communications, such as statements, images or instructions. I suggest "explicitly" is replaced with "clearly" or similar.

Figure 1g: are the replicates technical (measurements from the same cell batch on the same day) or biological (isolated from different animals)?

Figure 1h:

"1029 h, Poly(A) tails added to Ig κ mRNA can be removed by RNase H treatment in the presence of

1030 oligo(dT)20. High-resolution northern blot analysis of Ig κ mRNA and U6 RNA isolated from Tent5C

1031 KO (lane 1 and 3) and WT (lane 2 and 4) B cells activated with LPS and IL-4 for 7 days."

The removal of poly(A) tails is a key control which I'm thankful for, but this is not the essential point of the panel (I suggest a change in description). It should show a reduction in poly(A) tail size upon knock out. I think this is probably the case but it is very hard to see because of the differences in intensity. Could you change the exposure or overlay scans of lanes 1 and 2, for instance normalised to equal area under the curve?

Figure 1i: variability is quite large and it is not clear how the statistically significant difference (which is stated in the text) between WT and KO for Ig λ and κ was determined or what the half-life was. There appears to be no difference for the J chain. It is also not clear if the 6-8 replicates are technical replicates from one large experiment, or whether these are biological replicates of cells from different animals on the same or different days. For biological replicates, time points from the same replicate are co-dependent and should not be treated as independent measurements. As the data in 1j are clear, this experiment could be omitted or be moved to the supplement, but more information is needed for a correct interpretation.

Figure 2j: are these biological or technical replicates? (see above)

Line 135: "60 in WT to 41 adenosines in " state whether this is the average or median size

Line 195: change "Finally, to assess, if the lack of TENT5C affects" to " Finally, to assess if the lack of TENT5C affects" To remove unnecessary comma and prevent association with backsides or donkeys ;-)

Line 332: "fractions" instead of "factions"

Line 372: "immunoglobulin transcripts have significantly longer poly(A) tails
Line 372 compared to other highly abundant mRNAs, like mRNAs encoding ribosomal proteins"
This is indeed interesting, but it is not clear from the data shown. Supplementary figure 1h shows a distribution of ribosomal protein mRNA poly(A) tails with a mode of about 50 which doesn't appear much different from the 60 average or median cited for Ig mRNAs. By eye the difference with Figure 1d is not much different. Clearer evidence is required to support this claim if it is to be made.

Line 373: I also think (as one of the other reviewers remarked) that the contrast with Lima et al is overstated, as they do not claim that every highly expressed mRNA has a short poly(A) tail, it can be debated what is highly expressed and a 60 nucleotide poly(A) tail can hardly be classed as long. I suggest this discussion is removed.

While it is clear that TENT5C is predominantly cytoplasmic, a caveat needs inserting in the discussion indicating that it is likely but not certain that TENT5C polyadenylation is cytoplasmic, as no evidence for a lack of difference in the nucleus is presented.

The legends of supplementary figures are probably best inserted below each figure as they will presumably not be included in the manuscript formatting.

Cornelia de Moor

Reviewer #3 (Remarks to the Author):

The authors have done a thorough job of addressing my concerns.

Minor points

Both new northern blots (Figures 1H and 8A) do not have associated quantification. A histogram of intensity down the lane for each lane would suffice, and it would help readers interpret the tail length shift.

NCOMMS-19-25185A **Point by point response to the reviewer's comments**

We thank the Reviewers for a positive opinion about our paper. We addressed all minor issues raised by Reviewers, edited the manuscript accordingly and added caveats to the final conclusions. Finally, the title has been changed. Below we provide a point by point response to the Reviewer's comments.

Reviewers' comments:

Reviewer #1 (Remarks to the Author):

In the revised version, biochemical part is very good, but author still did not address my concerns. Indeed, in the response letter, authors mentioned that they find it difficult to the observed phenotypes are not, to a large extent, related to TENT5C mediated stabilization of Ig mRNA.

Thank you for appreciating the biochemical part of our paper. We have added caveats to the final conclusions and we have changed the title in order to avoid overstatements. We are convinced that our manuscript describes an interesting new phenomenon relevant to the humoral immune response. Thus, we are quite surprised by such a negative opinion by the Reviewer.

Reviewer #2 (Remarks to the Author):

The manuscript is much improved and I'd like to thank the authors for their excellent work. The data from the catalytic mutant mouse are an especially good addition that adds to the impact of this paper. The work is nearly ready for publication in my opinion.

I have just a few minor comments that need further clarification:

In my opinion, the authors misuse the word "explicitly" in the abstract. While explicit does mean "clear, exact, unambiguous", it is only used to refer to human communications, such as statements, images or instructions. I suggest "explicitly" is replaced with "clearly" or similar.

This has been changed in the revised version of the manuscript.

Figure 1g: are the replicates technical (measurements from the same cell batch on the same day) or biological (isolated from different animals)?

These are biological replicates. Such information has been added to the figure caption.

Figure 1h:

"1029 h, Poly(A) tails added to Ig κ mRNA can be removed by RNase H treatment in the presence of 1030 oligo(dT)₂₀. High-resolution northern blot analysis of Ig κ mRNA and U6 RNA isolated from Tent5C

1031 KO (lane 1 and 3) and WT (lane 2 and 4) B cells activated with LPS and IL-4 for 7

days.”

The removal of poly(A) tails is a key control which I’m thankful for, but this is not the essential point of the panel (I suggest a change in description). It should show a reduction in poly(A) tail size upon knock out. I think this is probably the case but it is very hard to see because of the differences in intensity. Could you change the exposure or overlay scans of lanes 1 and 2, for instance normalised to equal area under the curve?

The description of Figure 1h has been changed. Differences in signal intensity between lane 1 and 2 reflect diminished steady-state of this transcript in KO cells. To make this result more clear exposure shown in figure 1h has been changed. Moreover, we have added a new panel in supplementary figure (1j) where densitometric histograms of northern blots (Figure 1h and 8a) are shown.

Figure 1i: variability is quite large and it is not clear how the statistically significant difference (which is stated in the text) between WT and KO for Ig λ and κ was determined or what the half-life was. There appears to be no difference for the J chain. It is also not clear if the 6-8 replicates are technical replicates from one large experiment, or whether these are biological replicates of cells from different animals on the same or different days. For biological replicates, time points from the same replicate are co-dependent and should not be treated as independent measurements. As the data in 1j are clear, this experiment could be omitted or moved to the supplement, but more information is needed for a correct interpretation.

We agree that data from 4sU labeling are much more clear than half-life estimation based on actinomycin D treatment. However, both experimental approaches revealed a decrease in mRNA half-life in *TENT5C* KO cells.

We decided to move the “actinomycin D” experiment to the supplement section of the revised manuscript and extensively re-write this section to include all requested information.

Figure 2j: are these biological or technical replicates? (see above)

These are biological replicates. Such information has been added to the figure description.

Line 135: “60 in WT to 41 adenosines in ” state whether this is the average or median size

This is the most frequent value (mode) observed, or, in the other terms, the peak value observed on the density plots (Fig. 1. b-d). This was indicated in the text.

Line 195: change “Finally, to asses, if the lack of *TENT5C* affects” to “ Finally, to assess if the lack of *TENT5C* affects” To remove unnecessary comma and prevent association with backsides or donkeys ;-)

This has been changed.

Line 332: “fractions” instead of “factions”

The misspelling was corrected.

Line 372: “immunoglobulin transcripts have significantly longer poly(A) tails compared to other highly abundant mRNAs, like mRNAs encoding ribosomal proteins” This is indeed interesting, but it is not clear from the data shown. Supplementary figure 1h shows a distribution of ribosomal protein mRNA poly(A) tails with a mode of about 50 which doesn’t appear much different from the 60 average or median cited for Ig mRNAs. By eye the difference with Figure 1d is not much different. Clearer evidence is required to support this claim if it is to be made.

We agree that on average a difference in the poly(A) length between immunoglobulin mRNAs and transcripts encoding ribosomal proteins is not dramatic. However, this is only the average which shows the simplified view of poly(A) lengths for a particular group of transcripts. When looking at individual mRNAs (please see plot below, removed from the original manuscript during the first round of revision) it is clear that the range of mean poly(A) lengths is completely different for both mRNAs classes, whereas both are characterized by high expression values.

Figure R1. Scatter plot showing the relation between mean poly(A) length (x-axis, obtained with Nanopore direct RNA sequencing for WT) and expression (y-axis, obtained with Illumina RNAseq), for each individual transcript. Immunoglobulins and ribosomal proteins transcripts are marked in orange and blue, respectively.

Line 373: I also think (as one of the other reviewers remarked) that the contrast with Lima et al is overstated, as they do not claim that every highly expressed mRNA has a short poly(A) tail, it can be debated what is highly expressed and a 60 nucleotide poly(A) tail can hardly be classed as long. I suggest this discussion is removed.

We agree that such interpretation might be an overstatement, therefore we decided to remove this paragraph/section from the discussion.

While it is clear that TENT5C is predominantly cytoplasmic, a caveat needs inserting in the discussion indicating that it is likely but not certain that TENT5C polyadenylation is cytoplasmic, as no evidence for a lack of difference in the nucleus is presented.

Thank you for this comment. Indeed, there is no clear evidence in the manuscript showing that nuclear polyadenylation of immunoglobulin mRNAs is not changed. Such caveat has been added to the discussion section.

The legends of supplementary figures are probably best inserted below each figure as they will presumably not be included in the manuscript formatting.

We agree this would be helpful for readers. The legends of Supplementary figures are inserted below each figure in the revised version of the manuscript.

Reviewer #3 (Remarks to the Author):

The authors have done a thorough job of addressing my concerns.

Minor points

Both new northern blots (Figures 1H and 8A) do not have associated quantification. A histogram of intensity down the lane for each lane would suffice, and it would help readers interpret the tail length shift.

Thank you for a positive opinion about our work. We agree that such histograms would be helpful for the readers. Such information is now added as a new panel in Supplementary figure 1j.

REVIEWERS' COMMENTS:

Reviewer #2 (Remarks to the Author):

I have no further concerns and thank the authors for addressing my queries.

REVIEWERS' COMMENTS response:

Reviewer #2 (Remarks to the Author):

I have no further concerns and thank the authors for addressing my queries.

Thank you for a positive opinion about our paper and for numerous previous suggestions that improve the quality of our manuscript.